# REMem: Reasoning with Episodic Memory in Language Agents

**Yiheng Shu**[1], **Saisri Padmaja Jonnalagedda**[2], **Xiang Gao**[2], **Bernal Jiménez Gutiérrez**[1]
**Weijian Qi**[1], **Kamalika Das**[2], **Huan Sun**[1], **Yu Su**[1]
[1]The Ohio State University     [2]Intuit AI Research
{shu.251, su.809}@osu.edu, {saisri_jonnalagedda, xiang_gao}@intuit.com

## Abstract

Humans excel at remembering concrete experiences along spatiotemporal contexts and performing reasoning across those events, i.e., the capacity for episodic memory. In contrast, memory in language agents remains mainly semantic, and current agents are not yet capable of effectively recollecting and reasoning over interaction histories. We identify and formalize the core challenges of *episodic recollection* and *reasoning* from this gap, and observe that existing work often overlooks episodicity, lacks explicit event modeling, or overemphasizes simple retrieval rather than complex reasoning. We present REMem, a two-phase framework for constructing and reasoning with episodic memory: 1) *indexing*, where REMem converts experiences into a hybrid memory graph that flexibly links time-aware gists and facts. 2) *agentic inference*, where REMem employs an agentic retriever with carefully curated tools for iterative retrieval over the memory graph. Comprehensive evaluation across four episodic memory benchmarks shows that REMem substantially outperforms state-of-the-art memory systems such as Mem0 and HippoRAG 2, showing 3.4% and 13.4% absolute improvements on episodic recollection and reasoning tasks, respectively. Moreover, REMem also demonstrates more robust refusal behavior for unanswerable questions.[1]

## 1 Introduction

The ability to intentionally and precisely re-experience events from our past is one of the defining features of human intelligence. This "*mental time travel*" capacity, or *episodic memory*, allows us to access specific events along a spatiotemporal axis, specifying their order, duration, and even causality. Unlike *semantic memory*, which stores concepts and knowledge about the world, episodic memory shapes each individual's unique experiences and preferences, establishing a cornerstone for continual learning within one's environment.

Despite its importance for human cognition and rising interest in memory systems for language agents, achieving human-level episodic memory remains elusive, mainly due to the dominance of semantic memory paradigms in current approaches. Parametric memory, embedded in model weights during pre-training or fine-tuning, lacks adaptability and contextual grounding in specific experiences. Model editing methods (Yao et al., 2023b) can update stored facts but remain limited to modifying static semantic knowledge. Retrieval-augmented generation (RAG) systems with embedding models (Zhang et al., 2025; Lee et al., 2025a) enable dynamic knowledge access but still operate in a de-contextualized manner, divorced from spatiotemporal context. Finally, more advanced non-parametric systems, which use large language models (LLMs) to construct summaries or semantic graphs (Gutierrez et al., 2024; Gutiérrez et al., 2025; Edge et al., 2024), improve over RAG but still prioritize structured world knowledge over lived, interaction-specific experience.

Recently, a few studies have targeted episodic settings more directly. These typically represent episodic memory using entity relationships in (temporal) knowledge graphs with a loss of coherent event contexts (Rasmussen et al., 2025) or by selectively inferring what seems important and offering it as a summary (Chhikara et al., 2025; Tan et al., 2025), which lacks *explicit event modeling*. Critically, they fail to integrate situational dimensions, such as time, location, and participants, within

---

[1]Code and data are available at `https://github.com/intuit-ai-research/REMem`.

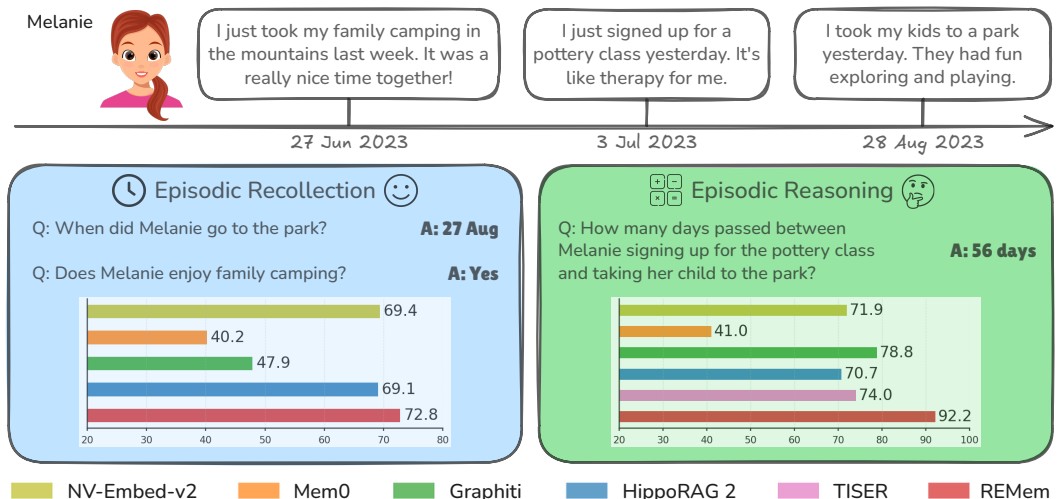

Figure 1: Overview of evaluation on episodic memory. Utterances are grounded to a timeline (top). We evaluate two progressive capabilities and show average scores on each (bottom): 1) Episodic recollection: recollect temporal and other situational elements of past experiences, measured by LLM-as-a-judge scores on LoCoMo and REALTALK. 2) Episodic reasoning: multi-step reasoning across the timeline based on recollection, e.g., event-to-event relations, counting, and ordinal queries, measured by LLM-as-a-judge score on Complex-TR and the EM score on Test of Time.

interaction histories. Furthermore, when memory must support reasoning across multiple linked events, existing methods rely heavily on similarity-based retrieval, offering limited capacity to infer complex inter-event relationships. Given these observations, we believe that a comprehensive event representation, combined with flexible retrieval and reasoning, deserves further exploration.

To advance episodic memory and reasoning for language agents, we first identify two key, progressive challenges, as shown in Figure 1: 1) **Episodic recollection**: reconstruct events and their situational dimensions based on experiences, such as time, location, participant, emotion; i.e., the ability to bind situational elements to specific events. 2) **Episodic reasoning**: multi-step reasoning based on episodic recollection, such as inter-event relations, ordinal constraints, and superlatives.

Then, we introduce a new episodic memory framework, REMem (Reasoning with Episodic Memory), for language agents. We formalize episodic memory as time-aware event representations and propose a *hybrid memory graph* that stores **gists** (concise, human-readable event summaries with parsed timestamps) and **facts** (time-scoped triples). Unlike existing work that selectively extracts useful information, we clearly *instruct* the LLM to construct memories organized primarily along time and linked to situational dimensions such as participants, locations, and emotions. We develop an *agentic inference* procedure with carefully curated tools for retrieval and graph exploration. As a result, memory management goes beyond simply matching isolated text spans and enables complex logical composition, including time-range filtering, neighbor exploration, and ordinal constraints.

We conduct one of the most comprehensive evaluations on episodic memory to date, covering four benchmarks on conversational and temporal reading comprehension. REMem delivers consistent gains over the current state of the art, achieving a $3.4\%$ and $13.4\%$ absolute improvement on episodic recollection and reasoning tasks, respectively. REMem also shows unparalleled reasoning capabilities, being the only method to exceed $90\%$ exact match score on the Test of Time benchmark (Fatemi et al., 2025), and behaves more robustly to refuse unanswerable questions. These strong results position REMem as a solid step towards bringing effective episodic memory to language agents.

## 2 RELATED WORK

### 2.1 NON-PARAMETRIC MEMORY FOR LLMS

A broad line of work augments LLMs with non-parametric memory. ChatGPT (OpenAI, 2025a) combines prior chats with user-controlled saved memories for personalization. Strong embedding

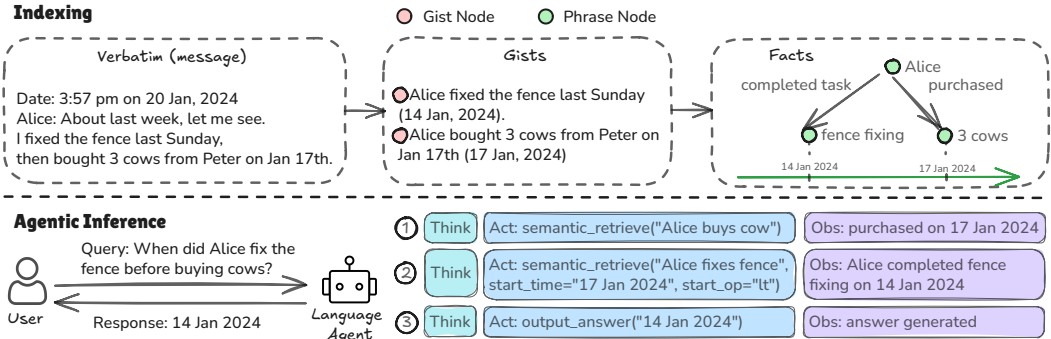

Figure 2: Overview of REMem. The indexing phase turns utterances into time-aware memory by extracting event gists and time-scoped facts (triples) and organizing them as a hybrid graph. The agentic inference phase invokes carefully curated tools over this graph to surface the most relevant gists and facts for reasoning in an iterative manner.

models (Lee et al., 2025a; Zhang et al., 2025) yield competitive retrieval, but their flat vector spaces do not explicitly encode episodic or temporal structure. Structure-augmented approaches build graphs or memory layers to improve multi-hop and cross-session retrieval (Edge et al., 2024): HippoRAG 1&2 (Gutierrez et al., 2024; Gutiérrez et al., 2025) organizes knowledge for associative retrieval and continual updates. Graphiti/Zep (Rasmussen et al., 2025) maintains temporal knowledge graphs and context-assembly pipelines. Mem0 (Chhikara et al., 2025) extracts and consolidates conversational memory in a graph structure with a knowledge update mechanism. Orthogonal to graph layering, MemGPT (Packer et al., 2023) implements OS-style hierarchical paging and virtual context management. A-Mem (Xu et al., 2025) performs agentic, Zettelkasten-like dynamic linking among notes. MemoryBank (Zhong et al., 2024) introduces time-decay–based consolidation and forgetting. Reflective Memory Management (Tan et al., 2025) further refines what to store and retrieve over time. Overall, most existing systems rely on LLM-based summarization or graph construction to extract key information, yet they overlook the crucial role of spatiotemporal context and situational elements in supporting episodic memory. These systems rarely propose an explicit design for episodic memory and reasoning. In contrast, our approach explicitly represents event gists and temporal facts grounded to a timeline, links them to core situational dimensions, and integrates them with tool-augmented reasoning.

## 2.2 EPISODIC MEMORY AND REASONING

Recent benchmarks increasingly stress long-horizon interaction for language agents: LoCoMo (Maharana et al., 2024) and REALTALK (Lee et al., 2025b) evaluate multi-session conversational memory in synthetic and real settings. Conversation understanding of LongMemEval (Wu et al., 2025) requires information extraction or multi-session reasoning, but most tasks can be seen as single-retrieval problems only requiring agents to retrieve the correct segments (Hu et al., 2025b). Temporal reading comprehension benchmarks, such as TimeQA (Chen et al., 2021), MenatQA (Wei et al., 2023), Complex-TR (Tan et al., 2024), and Test of Time (Fatemi et al., 2025), decompose temporal reasoning into diverse skills: event ordering, date resolution, counting, duration estimation, and timeline construction. Methodologically, timeline self-reflection method like TISER (Bazaga et al., 2025) improves temporal reasoning only through better prompting, while tool-use agents (Ge et al., 2025; Hu et al., 2025a) enable procedural interaction with non-parametric memory. Our work complements these efforts by 1) encoding episodes as time-aware gists and facts in a hybrid graph, and 2) using tool-augmented retrieval and graph exploration to perform episodic recollection and reasoning during inference.

## 3 METHODOLOGY

Our framework, REMem, allows language agents to reason with episodic memory through a two-phase process: *indexing* and *agentic inference*. The indexing phase builds a memory graph that

Table 1: Curated tools and their signature. Both the retrieval and graph exploration tools output two sets of results: a list of gists and a list of facts. See Appendix D for prompts and demonstrations.

| Type | Name | Arguments |
|------|------|-----------|
| Retrieval | semantic_retrieve | query, start_time, end_time, start_operator, end_operator |
| | lexical_retrieve | query, start_time, end_time, start_operator, end_operator |
| Graph Exploration | find_gist_contexts | gist_id, start_time, end_time, start_operator, end_operator |
| | find_entity_contexts | subject, object, predicate, start_time, end_time, start_operator, end_operator, limit, ordering, offset, aggregation |
| Flow Control | output_answer | answer |

stores the gists and facts for episodes. Then, the agentic inference phase flexibly leverages carefully curated tools to reason over user queries and retrieve from the memory graph.

## 3.1 INDEXING

Cognitive science reminds us that humans rely more on gist than verbatim memory when making decisions (Reyna & Brainerd, 1995), and our discourse memory is shaped around situation models, which integrate temporal, spatial, causal, and situational dimensions beyond surface details (Johnson-Laird, 1983; Zwaan & Radvansky, 1998). Besides, recent RAG studies show that hybrid memory structures combining concept-level and context-level information are more effective than either alone (Edge et al., 2024; Gutiérrez et al., 2025). Motivated by these insights, our principles for constructing a memory graph (Figure 2) focus on comprehensiveness and flexibility, which jointly encode gists with multiple situational dimensions and facts with temporal contexts. Specifically, we perform the following steps.

**1) Gist Extraction**: For each event statement or chat session, we generate one or multiple gist statements in natural language. Each gist is prefixed with the episode's timestamp (reference time) if applicable, and any relative temporal expressions are resolved to absolute dates. For each episode (e.g., a chat session), this yields a gist list, where each gist is a concise sentence capturing the episode's key details, including participants, actions, objects, locations, intentions, and quantities, in a single atomic event description. **2) Fact Extraction**: We further extract structured facts from each episode's text and the extracted gist list. These facts are represented as (subject, predicate, object) triples, where each field is a schemaless phrase, mainly capturing who did what to whom. We also extract temporal contexts (dates/times) and ground each fact in a timeline by optionally attaching Wikidata-style qualifiers (Vrandecic & Krötzsch, 2014), *point_in_time*, *start_time*, and *end_time*, to the corresponding triple. We preserve gists and facts as they are added over time, even when they are potentially contradictory, thereby maintaining a record that can be revisited historically. **3) Graph Construction**: Using the above outputs, we construct a memory graph that integrates gist nodes and phrase nodes. Gist nodes serve as *context-level* episodic representations, each connected to the phrase nodes extracted from the same chunk. At the *concept-level*, subject and object phrase nodes of each fact are directly linked by edges, encoding the extracted relationships between those phrases. Thus, a *hybrid* memory graph combining concept and context levels is formed. To further enhance connectivity, we add synonymy edges between gist nodes whose embedding similarity exceeds a threshold, following the HippoRAG 2 approach (Gutiérrez et al., 2025). This mechanism clusters semantically related gists (e.g., different phrasings of similar events) and enriches the graph with higher-level semantic connections.

## 3.2 AGENTIC INFERENCE

In contrast to the common RAG approach of simply performing a one-time matching with related text, our inference phase includes *iterative retrieval* to handle complex logic, and *mental time travel*, which filters memory entries according to temporal conditions. We adopt a ReAct-style agent (Yao et al., 2023a; Gu et al., 2024) equipped with three categories of carefully curated tools to access the hybrid graph: 1) retrieval tools, 2) graph exploration tools, and 3) the flow control tool. The results returned by retrieval or graph exploration tools contain both gists and facts to provide a

Table 2: The statistics of sampled datasets.

| Datasets | LoCoMo | REALTALK | Complex-TR | Test of Time |
|---|---|---|---|---|
| # of queries | 1,986 | 728 | 1,000 | 2,800 |
| # of chunks/messages | 5,882 | 8,944 | 1,095 | 124,919 |
| # of graphs | 10 | 10 | 1 | 560 |

comprehensive view. Specifically, the agent follows a three-stage protocol over the hybrid memory graph, as shown in Table 1. Full tool specifications are provided in Appendix D.

**1) Retrieval**: The agent first invokes *semantic_retrieve* (using an embedding model) or *lexical_retrieve* (using BM25) tool to obtain truncated seed nodes and their contexts from the graph, e.g., candidate entity IDs, temporal windows, and coarse topical cues. The agent decomposes complex questions into simpler sub-queries that guide the next step. **2) Graph Exploration**: Using the seed nodes retrieved from stage 1, the agent issues targeted calls to *find_gist_contexts* to obtain episode-level narratives and temporally grounded evidence, or uses *find_entity_contexts* when the query explicitly targets entities under a known graph schema. *find_entity_contexts* not only supports specifying subjects, predicates, or objects, but also filters gists or facts that meet specific temporal conditions. **3) Flow Control**: Once sufficient evidence has been gathered, the agent invokes *output_answer* to generate the final response upon reaching confidence. This procedure exploits the entire interaction history, incorporating both explored gists and facts, to conduct the concluding reasoning.

## 4 EXPERIMENTAL SETUP

### 4.1 DATASETS

For the episodic recollection task, we use conversational question answering (QA) benchmarks: **LoCoMo** (Maharana et al., 2024) is a synthesized conversational benchmark, while **REALTALK** (Lee et al., 2025b) is collected from real human conversations. Both benchmarks include questions that cover temporal aspects and non-temporal situational aspects, and we use all samples from them. For the episodic reasoning task, we use reading comprehension benchmarks **Complex-TR** (Tan et al., 2024) and **Test of Time** (Fatemi et al., 2025). From Complex-TR, we randomly sample 1,000 queries. From the Test of Time, we adopt all samples from its semantic part, since this part explicitly requires memory.

### 4.2 METHODS FOR COMPARISON

We select state-of-the-art methods from the following categories for comparison: 1) Strong embedding models from the MTEB benchmark (Muennighoff et al., 2023) for RAG setting, including **Qwen/Qwen3-Embedding-8B** (Zhang et al., 2025) and **nvidia/NV-Embed-v2** (Lee et al., 2025a). 2) Structure-augmented memory approaches, including **Mem0** (Chhikara et al., 2025), **Graphiti** (Rasmussen et al., 2025), and **HippoRAG 2** (Gutiérrez et al., 2025), where the first two methods are originally evaluated on episodic memory tasks. We reproduced the experiments using their *open-source* implementation rather than proprietary ones. 3) The prompt method **TISER** for temporal reasoning (Bazaga et al., 2025): give the query and retrieved contexts, we use its prompt for the final generation. The method is orthogonal to any memory system. 4) Additional references: **Oracle Message** takes the oracle retrieval results and only performs generation. **Full-Context** uses the entire corpus and the query for generation. Due to its limited context window, it should not be regarded as a memory method, but we take it as a reference rather than a comparative memory method.

### 4.3 METRICS

For Test of Time (Fatemi et al., 2025), we use the unique metric exact match (**EM**) score. For the remaining benchmarks, we use token-based **F1**, **BLEU-1** (Papineni et al., 2002), and **LLM-as-a-judge** scores (denoted as LLM-J later) as metrics for QA tasks. We adopt the F1 calculation from HippoRAG 2 (Gutiérrez et al., 2025), BLEU-1 implementation from HuggingFace Evaluate

Table 3: Performance (%) on episodic recollection task. The highest value and second-highest value in each column are bold and underlined, respectively. Numbers are means with 95% bootstrap confidence intervals as subscripts and superscripts. The same applies to the tables below.

| Methods | LoCoMo (1, 986) | | | REALTALK (728) | | |
|---|---|---|---|---|---|---|
| | F1 | BLEU-1 | LLM-J | F1 | BLEU-1 | LLM-J |
| Oracle Message | 48.0 | 38.3 | 81.0 | − | − | − |
| Full-Context | 37.8 | 28.6 | 76.7 | 25.3 | 18.6 | 65.1 |
| *Large Embedding Models* | | | | | | |
| Qwen3-Embed-8B | $35.3^{+1.7}_{-1.6}$ | $28.9^{+1.8}_{-1.5}$ | $64.2^{+2.4}_{-2.0}$ | $20.2^{+1.8}_{-1.6}$ | $14.9^{+1.6}_{-1.4}$ | $52.5^{+3.4}_{-3.6}$ |
| NV-Embed-v2 (7B) | $39.6^{+1.7}_{-1.4}$ | $31.0^{+1.7}_{-1.4}$ | $73.0^{+2.0}_{-1.8}$ | $23.8^{+1.9}_{-1.8}$ | $17.7^{+1.5}_{-1.5}$ | $59.5^{+3.3}_{-3.6}$ |
| *Structure-Augmented Memory* | | | | | | |
| Mem0 | $25.1^{+1.7}_{-1.5}$ | $18.0^{+1.4}_{-1.1}$ | $49.7^{+2.3}_{-2.2}$ | $9.8^{+1.5}_{-1.3}$ | $7.2^{+1.1}_{-1.0}$ | $14.3^{+2.7}_{-2.3}$ |
| Graphiti | $33.7^{+1.8}_{-1.8}$ | $28.9^{+1.9}_{-1.7}$ | $52.5^{+2.3}_{-2.3}$ | $15.1^{+1.8}_{-1.5}$ | $11.5^{+1.4}_{-1.2}$ | $35.3^{+3.7}_{-3.3}$ |
| HippoRAG 2 | $39.0^{+1.6}_{-1.6}$ | $30.8^{+1.5}_{-1.5}$ | $74.0^{+1.7}_{-2.1}$ | $21.9^{+1.6}_{-1.6}$ | $16.2^{+1.4}_{-1.3}$ | $55.8^{+3.4}_{-3.6}$ |
| *Ours* | | | | | | |
| REMem-I | $\mathbf{42.4}^{+1.6}_{-1.6}$ | $\mathbf{32.7}^{+1.5}_{-1.6}$ | $\underline{76.2}^{+2.0}_{-1.9}$ | $\underline{25.6}^{+1.8}_{-1.4}$ | $\underline{18.1}^{+1.6}_{-1.4}$ | $\underline{63.7}^{+3.6}_{-3.5}$ |
| REMem-S | $\underline{41.3}^{+1.6}_{-1.5}$ | $\underline{31.5}^{+1.6}_{-1.4}$ | $\mathbf{77.5}^{+1.9}_{-1.6}$ | $\mathbf{26.2}^{+1.8}_{-1.6}$ | $\mathbf{19.2}^{+1.5}_{-1.3}$ | $\mathbf{65.3}^{+3.6}_{-3.1}$ |

Table 4: Performance (%) on episodic reasoning tasks.

| Methods | Complex-TR (1, 000) | | | Test of Time (2, 800) |
|---|---|---|---|---|
| | F1 | BLEU-1 | LLM-J | EM |
| Full-Context | 74.2 | 68.0 | 81.6 | 79.7 |
| *Large Embedding Models* | | | | |
| Qwen3-Embed-8B | $77.1^{+2.3}_{-2.1}$ | $71.4^{+2.5}_{-2.4}$ | $80.9^{+2.5}_{-2.5}$ | $70.3^{+1.8}_{-1.7}$ |
| NV-Embed-v2 (7B) | $77.5^{+2.2}_{-2.1}$ | $71.9^{+2.3}_{-2.3}$ | $80.4^{+2.6}_{-2.5}$ | $68.9^{+1.7}_{-1.7}$ |
| w/ TISER | $\underline{88.1}^{+1.7}_{-1.5}$ | $\underline{83.6}^{+2.2}_{-1.8}$ | $88.3^{+1.9}_{-1.8}$ | $68.9^{+1.8}_{-1.8}$ |
| *Structure-Augmented Memory* | | | | |
| Mem0 | $43.1^{+2.8}_{-2.7}$ | $35.1^{+2.5}_{-2.4}$ | $41.0^{+3.0}_{-3.0}$ | − |
| Graphiti | $76.6^{+2.2}_{-2.3}$ | $71.4^{+2.4}_{-2.5}$ | $78.8^{+2.6}_{-2.6}$ | − |
| HippoRAG 2 | $78.2^{+2.3}_{-1.8}$ | $72.7^{+2.4}_{-2.4}$ | $81.5^{+3.4}_{-1.3}$ | $66.9^{+1.7}_{-1.7}$ |
| *Ours* | | | | |
| REMem-I | $83.3^{+1.8}_{-1.8}$ | $77.6^{+2.2}_{-2.1}$ | $\underline{89.6}^{+2.0}_{-2.0}$ | $\mathbf{93.1}^{+0.9}_{-1.1}$ |
| w/ TISER | $\mathbf{90.6}^{+1.2}_{-1.4}$ | $\mathbf{86.0}^{+1.7}_{-1.7}$ | $\mathbf{92.0}^{+1.6}_{-1.7}$ | $\underline{90.6}^{+1.0}_{-1.2}$ |
| REMem-S | $78.5^{+2.0}_{-2.1}$ | $72.7^{+2.4}_{-2.4}$ | $82.6^{+2.3}_{-2.4}$ | $72.5^{+1.8}_{-1.8}$ |

(HuggingFace, 2025). We adopt the same LLM evaluation prompts as Mem0 (Chhikara et al., 2025), which account for both temporal and conversational contexts.

## 4.4 IMPLEMENTATION DETAILS

We use GPT-4.1-mini-2025-04-14 (OpenAI, 2025b) as the default LLM and nvidia/NV-Embed-v2 (Lee et al., 2025a) as the default embedding model for both extraction and QA tasks, in REMem as well as in comparison methods. For baselines using the embedding model, we retrieve the top-10 original passages (messages in conversation). For Mem0 and Graphiti, we retrieve the top-10 of their processed chunks. REMem applies the same retrieval scope for each step, operating over the top-10 gists and facts. We use the top-3 returned sessions from HippoRAG 2 for final generation.

We explore two settings for our method: **REMem-I**(terative) autonomously selects tools in a multi-step retrieval and reasoning process, following the protocol in §3.2. **REMem-S**(ingle) only adopts a single-step embedding-based retrieval and then generation. The maximum number of agentic inference steps in REMem-I for each dataset is selected from 2 to 5 based on a small validation set. Then, we use 3 for episodic recollection tasks and 5 for episodic reasoning tasks. The similarity threshold for synonymy edges is set to 0.8, following HippoRAG 2.

Table 5: Ablation study on LoCoMo and Complex-TR, regarding the graph structure and the usage of retrieval tools.

| Methods | LoCoMo | | | Complex-TR | | |
|---|---|---|---|---|---|---|
| | F1 | BLEU-1 | LLM-J | F1 | BLEU-1 | LLM-J |
| REMem-S | 41.3 | 31.5 | **77.5** | 78.5 | 72.7 | 82.6 |
| REMem-I | **42.4** | 32.7 | 76.2 | **83.3** | **77.6** | **89.6** |
| w/o Gists | 31.7 | 28.7 | 48.9 | 80.3 | 75.9 | 80.9 |
| w/o Facts | 42.0 | 32.6 | 74.1 | 80.5 | 74.5 | 87.2 |
| w/o Synonymy edges | 37.6 | 28.7 | 76.4 | 81.6 | 75.6 | 89.2 |
| w/o Tool semantic_retrieve | 41.7 | **33.3** | 72.8 | 82.4 | 76.4 | 88.1 |
| w/o Tool lexical_retrieve | 40.6 | 31.2 | 76.8 | 81.7 | 75.8 | 87.5 |

## 5 RESULTS

### 5.1 EPISODIC RECOLLECTION

The results on episodic recollection tasks are shown in Table 3. Overall, REALTALK collected from human utterances poses greater challenges than synthetic LoCoMo. As a reference, Full-Context performs well, but the inference cost is notable: the average number of input tokens for each LoCoMo query is 26k, while REMem-I and REMem-S consume 9k and 0.9k tokens, respectively, during the inference phase. The RAG method using large embedding models serves as a strong baseline, especially NV-Embed-v2. Structure-augmented memory methods demonstrate subpar overall performance, particularly on REALTALK, where human utterances are more spontaneous, noisy, and less dense in information. Mem0 extracts many statements, but most are subsequently discarded from memory by its own decision, resulting in only a few details being remembered. Graphiti constructs a temporal knowledge graph centered around entities, which results in the loss of coherent contextual information related to various situational dimensions of events. HippoRAG 2 lacks any modeling of the temporal dimension or events. Its overall performance is only comparable to the embedding baseline due to its use of passage nodes with embedding-based retrieval. In contrast, REMem-I and REMem-S substantially outperform existing methods and even approach the oracle performance. More detailed results for LoCoMo and REALTALK are provided in Appendix C.1 and C.2, respectively, where we show REMem-I and REMem-S have their respective strengths across different metrics. Notably, cross-session questions in LoCoMo account for only $14.2\%$ of the benchmark, and support messages for most questions originate from a single chat session, which explains why REMem-S as a single-step variant performs better in certain scenarios.

### 5.2 EPISODIC REASONING

The results on episodic reasoning tasks are shown in Table 4. RAG using large embedding models remains a strong baseline. TISER, as a prompt guiding language agents to perform temporal reasoning, demonstrates a stronger reasoning capability on Complex-TR, compared to our straightforward answer-generation prompts (Figure 6). However, it remains a fixed prompt that struggles to cover all episodic reasoning challenges, and excels at chronological questions, such as before/after or first/last types. The structure-augmented memory methods mainly leverage embedding to retrieve at various granularities, e.g., entities or summaries. But this simple semantic matching is insufficient for capturing the full contextual information necessary for complex reasoning, and it fails to support the required logical operations. REMem shows absolute superiority in complex reasoning tasks. In particular, REMem-I demonstrates a clear advantage over REMem-S (LLM-J +7.0, EM +20.6) and becomes the **unique one exceeding** $90\%$ **EM score**, benefiting from multi-step retrieval and flexible tool use. Moreover, REMem-I obtains substantially larger improvements over Full-Context (LLM-J +8.0, EM +13.4) on these challenging episodic reasoning tasks, compared to that on episodic recollection tasks. More detailed results for Complex-TR and Test of Time are provided in Appendix C.3 and C.4, respectively.

Table 6: Performance of refusal on LoCoMo. Among $1,986$ queries, $446$ are unanswerable. Precision is the proportion of predicted unanswerable cases that are indeed unanswerable (correct refusals as "no information available"). Recall is the proportion of truly unanswerable cases correctly predicted as such. The F1 score is computed from these precision and recall values.

| Methods | # of Refusals | Precision (%) | Recall (%) | F1 (%) |
|---------|---------------|---------------|------------|--------|
| Graphiti | 954 | 38.9 | **83.6** | 53.1 |
| Mem0 | 90 | 40.0 | 8.1 | 13.5 |
| REMem | 344 | **73.3** | 56.8 | **64.0** |

## 6 DISCUSSIONS

### 6.1 ABLATION STUDY

We conducted ablation experiments on LoCoMo and Complex-TR (Table 5). Both gists and facts are important, but they contribute in different ways. Removing gists leads to the largest degradation, especially on LoCoMo, where LLM-J drops from 76.2 to 48.9. This supports our design choice that gists carry the main situational elements. Removing facts produces a smaller yet consistent drop, particularly on Complex-TR, where LLM-J decreases from 89.6 to 87.2. This suggests that facts play a supporting role for multi-hop reasoning, by providing concrete anchors that connect concepts across sessions (Appendix C.6). Graph structure and retrieval tools also matter. Ablating synonymy edges reduces F1 and BLEU-1 on both datasets, indicating that modeling synonymic relationships improves lexical robustness and recall, while LLM-J remains almost unchanged, which implies that the core reasoning path is largely preserved. Finally, removing either the semantic or the lexical retrieval tool hurts performance. Without semantic_retrieve, LLM-J on Complex-TR decreases from 89.6 to 88.1 and also drops on LoCoMo, showing that semantic retrieval is crucial for finding conceptually relevant memories. Without lexical_retrieve, F1 and BLEU-1 decline and LLM-J on Complex-TR falls to 87.5, indicating that lexical retrieval complements semantic retrieval by improving coverage of surface forms.

### 6.2 REFUSAL PERFORMANCE

In real-world applications, users will always pose queries for which the system lacks sufficient context. Such adversarial or unanswerable questions are still valid for evaluation purposes, and we include them as part of the full LoCoMo benchmark. For these cases, LoCoMo labels them as "*no information available*". Therefore, if a method either fails to produce an answer or explicitly outputs this phrase as instructed, we take it as a refusal to answer. For these metrics of refusal behaviors (not QA metrics) as shown in Table 6, REMem achieves the highest F1 (63.96%) by coupling the best precision (73.3%) with competitive recall (56.8%). Compared to Graphiti, which produces 954 refusals with low precision (38.9%) despite the highest recall (83.6%), REMem improves precision by +34.4 points and F1 by +10.9 points, while producing roughly one-third as many refusals (344 vs. 954), indicating substantially fewer unnecessary rejections. Compared to Mem0, which is overly permissive (only 90 refusals), REMem correctly flags more unanswerable cases. Overall, REMem achieves a better balance for refusal behavior on adversarial questions.

### 6.3 HUMAN EVALUATION

To demonstrate the validity of using LLM as a judge in episodic memory, especially for reasoning tasks, we randomly selected 100 samples from LoCoMo and conducted a human evaluation (Table 7). Both the LLM judge and the human evaluation use binary scores. The LLM judge matched human scores in 93% of cases, with only 7 discrepancies. Five LLM-accepted answers were manually judged as incorrect due to incomplete lists or temporal reasoning errors. Conversely, two LLM-rejected answers were manually judged as correct, because valid paraphrases were not recognized. Though some limitations remain, these findings suggest that using LLM as a judge produces evaluations most closely aligned with human judgment compared to traditional metrics.

Table 7: Comparison between automatic metrics and human evaluation. Mean denotes the mean value over 100 selected LoCoMo samples. The correlations are compared with human evaluation.

| Metric | Mean | Pearson $r$ w/ Human | Spearman $\rho$ w/ Human |
|---|---|---|---|
| Human Evaluation | 0.710 | – | – |
| F1 | 0.410 | 0.551 | 0.603 |
| BLEU-1 | 0.284 | 0.417 | 0.531 |
| LLM Judge | 0.740 | **0.827** | **0.827** |

## 6.4 ERROR ANALYSIS

We conduct an error analysis for REMem on LoCoMo and Complex-TR. In 100 sampled LoCoMo errors, **the most common are selection or grounding errors** ($46\%$), where REMem locates the correct or similar slot but assigns the wrong value or misinterprets the referent in details. For example, when asked "*What is Nate's favorite video game?*" REMem responded "*Catan*", but that was only one of Nate's interests rather than his favorite "*Xenoblade Chronicles*". **Temporal or numerical reasoning mistakes** ($19\%$) include errors in relative dates or durations. For instance, to the question "*When is Nate hosting a gaming party?*" the model answered "*18–19 June 2022*" while the gold reference was "*the weekend after 3 June 2022*". **Another 18% are abstentions despite the evidence being retrieved**, where the model incorrectly claims that no information is available, even though the gold answer is present. In 100 sampled Complex-TR errors, the most frequent failure mode ($42\%$) was **temporal window mismatch**, where correct entities are retrieved but misaligned with the specified time span. For instance, when asked "*Which employers did Ott-Heinrich Keller work for from Aug 1941 to Mar 1945?*" the gold answer included both "*the Naval Academy at Mürwik*" and "*the University of Münster*", but our prediction listed only the first. Roughly $21\%$ of cases were due to **incomplete or inconsistent multi-entity lists**, where some retrieved items are missing or extraneous. About $18\%$ are **offset direction mistakes**, such as confusing *before* and *after* or skipping to a later hop. For the question "*Which employer did Karyn A. Temple work for after RIAA?*" where the gold answer was "*the U.S. Department of Justice*" but the model instead gave "*the Copyright Office*". A smaller fraction ($\approx 5\%$) of ours incorrectly returns "*no information available*" despite gold facts being present.

## 6.5 COMPARATIVE ANALYSIS: REMEM VS. RAG

We provide a few qualitative examples to compare REMem with a RAG system using NV-Embed-v2. REMem outperforms NV-Embed-v2 on questions that require disambiguation across brand categories and reconciliation of time-stamped events, while the embedding baseline shines on straightforward interval calculations. For Q1 in Table 8, REMem extracts and reasons over a normalized gist, surfacing an explicit dated summary ("*[19 Dec 2023] John got an amazing deal with a renowned outdoor gear company*"), and therefore matches the gold label exactly. NV-Embed-v2 selects a semantically similar but incorrect brand vertical (beverage). For Q2, REMem correctly chains events and compares timestamps (Susie was adopted around Aug 2021, while Seraphim was adopted last year), whereas the embedding method fixates on an earlier mention and ignores the newer adoption. Conversely, for the comparative Q3, NV-Embed-v2 retrieves both dates cleanly and computes the three-month interval, while REMem misaligns events in a longer context, yielding a one-month error. Overall, these examples indicate that time-aware gist extraction and agentic inference provide generally better robustness to distractors and temporal ambiguity, whereas a common RAG pipeline can be reliable when the answer reduces to simple computation over accurately retrieved facts. An additional comparative analysis between REMem and TISER on Complex-TR can be found in Appendix F.1.

## 7 CONCLUSION

Challenges like episodic recollection and reasoning are far from solved for language agents. We presented REMem, a time-aware episodic memory framework. The proposed hybrid memory graph unifies concept-level and context-level information with flexible temporal awareness, while the agentic retriever enables integration of retrieval and reasoning. REMem consistently demonstrates better performance on episodic recollection and reasoning tasks across four benchmarks, with better

Table 8: Comparison between NV-Embed-v2 and REMem on LoCoMo. T or F indicates the LLM's judgment (True or False).

| Question | Gold Answer | NV-Embed-v2 | REMem |
| --- | --- | --- | --- |
| Q1: What kind of deal did John get in December? | Deal with an outdoor gear company | Endorsement deal with a beverage company (F) | Endorsement deal with an outdoor gear company (T) |
| Q2: Which pet did Jolene adopt more recently, Susie or Seraphim? | Seraphim | Susie (F) | Seraphim (T) |
| Q3: How many months passed between Andrew adopting Toby and Buddy? | Three months | 3 months (T) | 1 month (F) |

refusal behavior for unanswerable queries and higher token efficiency. REMem indicates a promising step toward more reliable long-horizon language agents. Future work should consider long-term memory for language agents operating in more complex environments. Building memories in a streaming format also poses an engineering challenge compared to offline batch indexing.

## ACKNOWLEDGMENTS

The authors would like to thank colleagues from the OSU NLP group and Intuit AI Research for constructive discussions. This material is based upon work supported by the National Science Foundation under Grant No. 2443149. Any opinions, findings, and conclusions or recommendations expressed in this material are those of the author(s) and do not necessarily reflect the views of the National Science Foundation. This work is also supported by an Alfred P. Sloan Foundation Fellowship.

## REPRODUCIBILITY STATEMENT

We have provided the implementation details for reproducing the method and experiments. All datasets used in this paper are publicly available, and we present dataset sampling in §4.1. The metrics calculation is shown in §4.3. The implementation details of REMem and the methods for comparison are presented in §4.4. The details on Mem0 and Graphiti are further provided in Appendix E.

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

# Appendix Contents

## A  THE USE OF LARGE LANGUAGE MODELS

Large language models did not play a significant role in the ideation or writing of this paper. They were only used for grammar checking and minor polishing of individual sentences.

## B  FORMAL DEFINITION

Let $\mathcal{M} = (V, E)$ be a typed multigraph with $V = V_{\text{gist}} \cup V_{\text{phrase}}$ and $E = E_{\text{rel}} \cup E_{\text{ctx}} \cup E_{\text{syn}}$. The graph thus contains two node types and three edge types.

**Node**  Each *gist node* $g \in V_{\text{gist}}$ stores a natural-language episode summary $\text{text}(g)$ and an optional time scope $\tau(g)$ (point or interval), and represents a context-level, human-readable description of an event that jointly encodes its situational dimensions, such as participants, actions, objects, locations, intentions, and quantities, anchored to that time scope. Each *phrase node* $p \in V_{\text{phrase}}$ stores a short string $\text{name}(p)$ and represents a concept-level element extracted from a factual triple (subject, predicate, object), typically denoting a participant, action, or object in the event. Phrase nodes can inherit temporal qualifiers such as `point_in_time`, `start_time`, or `end_time` from the underlying fact.

**Edge**  Each *relation edge* $e = (p_s, r, p_o, \tau(e)) \in E_{\text{rel}}$ links a subject phrase node $p_s$ and an object phrase node $p_o$ with a predicate $r$ and a validity interval $\tau(e)$, encoding fact-level relations such as

---

**Algorithm 1** Indexing$(\mathcal{D}; P_{\text{gist}}, P_{\text{fact}}, \theta_{\text{syn}})$

---

**Require:** Event statements $\mathcal{D} = \{d_i\}$ with timestamps when available; prompts for gist and fact extraction $P_{\text{gist}}, P_{\text{fact}}$; synonymy threshold $\theta_{\text{syn}}$.
**Ensure:** Typed memory graph $\mathcal{M}$ and retrieval indices.
1: Initialize $V_{\text{gist}}, V_{\text{phrase}}, E_{\text{rel}}, E_{\text{ctx}}, E_{\text{syn}} \leftarrow \emptyset$.
2: **for** $d \in \mathcal{D}$ **do**
3:     $\mathcal{G} \leftarrow \text{EXTRACTGISTS}(d; P_{\text{gist}})$            $\triangleright$ LLM returns a set of gists with text$(g)$ and $\tau(g)$
4:     **for** each $g \in \mathcal{G}$ **do**
5:        add $g$ to $V_{\text{gist}}$
6:     $\mathcal{F} \leftarrow \text{EXTRACTFACTS}(d; P_{\text{fact}})$             $\triangleright$ LLM returns set of $(p_s, r, p_o, \tau)$
7:     $P_d \leftarrow \emptyset$                                 $\triangleright$ phrase nodes appearing in this $d$
8:     **for** each $(p_s, r, p_o, \tau) \in \mathcal{F}$ **do**
9:        add $p_s, p_o$ to $V_{\text{phrase}}$ if new
10:       add $e = (p_s, r, p_o, \tau)$ to $E_{\text{rel}}$
11:       add $p_s, p_o$ to $P_d$
12:     **for** each $g \in \mathcal{G}$ **do**
13:        **for** each $p \in P_d$ **do**
14:           add $(g, p)$ to $E_{\text{ctx}}$         $\triangleright$ bind all gists to all phrases from the same $d$
15: **for all** pairs $(u, v)$ in $V_{\text{gist}}$ **do**
16:     **if** $\text{SIM}(u, v) \geq \theta_{\text{syn}}$ **then**
17:        add $(u, v)$ to $E_{\text{syn}}$
18: Build embedding/BM25 indices over $V$ and $\tau(\cdot)$
19: **return** $\mathcal{M} = (V, E)$ with indices

---

"Alice —bought→ cows" together with their temporal scope derived from these qualifiers. Each *context edge* $e = (g, p) \in E_{\text{ctx}}$ connects a gist node $g$ to a phrase node $p$ extracted from the same source chunk that gave rise to $g$, thereby linking an abstract episode summary (e.g., "Alice bought cows on Jan. 17th.") to its underlying factual triple $(\text{Alice}, \text{bought}, \text{cows})$ and associated temporal qualifier (e.g., `point_in_time` = "Jan. 17th"). Each *synonymy edge* $e = (g_i, g_j) \in E_{\text{syn}}$ links two gist nodes whose text embeddings exceed a similarity threshold, clustering semantically equivalent or highly related episodes and enriching the graph with associative links.

**Index** We maintain retrieval indices over this graph: an embedding index over text$(g)$ and name$(p)$, and a lexical (BM25) index over the surface forms of both gists and facts.

**Algorithm** The algorithmic procedures for indexing and agentic inference are shown in Algorithm 1 and 2, respectively.

## C    DETAILED RESULTS

### C.1    LOCOMO

The LoCoMo performance is presented in Table 9 and Table 10. For Oracle Message, while using annotations avoids retrieval challenges, QA remains a non-trivial task. Overall, large embedding models serve as strong baselines, and structure-augmented memory methods show inferior performance. Compared with NV-Embed-v2, HippoRAG 2 achieves only comparable LLM-J scores (+1.0). In contrast, REMem-I outperforms NV-Embed-v2 on both F1 and J scores (F1 +2.8, J +3.2), achieving performance most closely aligned with the oracle messages among all evaluated methods. REMem-S further improves the J score by 1.3 and attains the highest J scores on LoCoMo. We find that REMem-S outperforms REMem-I in many single-session settings, suggesting that most queries do not require complex multi-step retrieval, which can instead introduce contextual noise. A notable bias of this benchmark is that multi-session queries account for only 14.2% of all queries. Importantly, unlike previous studies, we also evaluate performance under adversarial settings, which we regard as essential for evaluating a system's refusal capability and mitigating hallucinations. A more detailed analysis of model refusal behavior is provided in §6.2. Full-Context achieves the

---

**Algorithm 2** AgenticInference($q, \mathcal{M}, T_{\max}$)

---

**Require:** Query $q$; hybrid memory graph $\mathcal{M}$ with indices; iteration limit $T_{\max}$

1:   $\mathcal{E} \leftarrow \emptyset$                 $\triangleright$ evidence: accumulated gists and facts

2:   $\mathcal{H} \leftarrow []$          $\triangleright$ interaction history (thoughts, actions, observations)

3:   $k \leftarrow 0$                  $\triangleright$ iteration counter

4:  **while** $k < T_{\max}$ **do**

5:     $t \leftarrow \text{LLM\_PLAN}(q, \mathcal{E}, \mathcal{H})$             $\triangleright$ reasoning step

6:     $a \leftarrow \text{LLM\_SELECTACTION}(t)$

7:     **if** $a$.name = `output_answer` **then**

8:         **return** $\text{LLM\_SYNTHESIZE}(q, \mathcal{E}, \mathcal{H})$     $\triangleright$ final answer based on gathered evidence

9:     **else if** $a$.name = `semantic_retrieve` **then**

10:       $(S, O) \leftarrow \text{SEMANTICRETRIEVE}(\mathcal{M}, a.\text{subquery})$     $\triangleright$ embedding search; returns seed nodes $S$ and observations $O$

11:       $\mathcal{E} \leftarrow \mathcal{E} \cup O; \mathcal{H} \leftarrow \mathcal{H} \,\|\, [t, a, O]$

12:     **else if** $a$.name = `lexical_retrieve` **then**

13:       $(S, O) \leftarrow \text{LEXICALRETRIEVE}(\mathcal{M}, a.\text{subquery})$             $\triangleright$ BM25 search

14:       $\mathcal{E} \leftarrow \mathcal{E} \cup O; \mathcal{H} \leftarrow \mathcal{H} \,\|\, [t, a, O]$

15:     **else if** $a$.name = `find_gist_contexts` **then**

16:       $O \leftarrow \text{FINDGISTCONTEXTS}(\mathcal{M}, a.\text{seeds}, a.\text{temporal\_constraints})$

17:       $\mathcal{E} \leftarrow \mathcal{E} \cup O; \mathcal{H} \leftarrow \mathcal{H} \,\|\, [t, a, O]$

18:     **else if** $a$.name = `find_entity_contexts` **then**

19:       $O \leftarrow \text{FINDENTITYCONTEXTS}(\mathcal{M}, a.\text{subject}, a.\text{predicate}, a.\text{object}, a.\text{temporal\_constraints})$    $\triangleright$ filter by phrase/predicate/temporal constraints

20:       $\mathcal{E} \leftarrow \mathcal{E} \cup O; \mathcal{H} \leftarrow \mathcal{H} \,\|\, [t, a, O]$

21:     $k \leftarrow k + 1$

22: **return** $\text{LLM\_SYNTHESIZE}(q, \mathcal{E}, \mathcal{H})$

---

best results on multi-session questions, which may suggest that this benchmark imposes only limited demands on long-context reasoning ability. Additional results on LoCoMo are also reported in Appendix C.1.

By default, we use the top-10 chunks on LoCoMo for embedding models. Notably, we also evaluate NV-Embed-v2 with only the top-3 chunks (messages) and find that it constitutes a strong baseline. Even under this constrained setting, it surpasses Qwen3-Embedding-8B and some structure-augmented memory systems, except for HippoRAG 2, which is configured to use the top-3 sessions.

Table 9: Performance (%) on LoCoMo (Part 1/2).

| Methods | Avg (1,986) | | | Single-Hop (321) | | | Multi-Hop (282) | | |
|---|---|---|---|---|---|---|---|---|---|
| | F1 | BLEU-1 | LLM-J | F1 | BLEU-1 | LLM-J | F1 | BLEU-1 | LLM-J |
| Oracle Message | 48.0 | 38.3 | 81.0 | 36.5 | 28.2 | 85.7 | 41.6 | 26.9 | 84.8 |
| Full-Context | 37.8 | 28.6 | 76.7 | 29.1 | 20.6 | 84.4 | 34.2 | 18.8 | 75.2 |
| *Large Embedding Models* | | | | | | | | | |
| Qwen3-Embed-8B | 35.3 | 28.9 | 64.2 | 24.8 | 18.1 | 64.2 | 24.6 | 15.1 | 57.1 |
| NV-Embed-v2 (7B) | 39.6 | 31.0 | 73.0 | 30.9 | _22.6_ | 77.9 | 29.4 | 17.8 | 64.9 |
|   w/ Top-3 Messages | 39.0 | _32.1_ | 67.2 | 26.5 | 20.2 | 68.2 | 21.1 | 13.6 | 51.4 |
| *Structure-Augmented Memory* | | | | | | | | | |
| Mem0 | 25.1 | 18.0 | 49.7 | 10.2 | 6.3 | 32.7 | 29.9 | 15.9 | 52.1 |
| Graphiti | 33.7 | 28.9 | 52.5 | 6.4 | 3.4 | 30.8 | 25.1 | 16.1 | 53.6 |
| HippoRAG 2 | 39.0 | 30.8 | 74.0 | 25.8 | 18.0 | 72.0 | 30.7 | **19.5** | **72.3** |
| *Ours* | | | | | | | | | |
| REMem-I | **42.4** | **32.7** | _76.2_ | **37.9** | **25.2** | _81.3_ | **32.6** | 18.7 | _70.2_ |
| REMem-S | _41.3_ | 31.5 | **77.5** | _35.4_ | 22.6 | **86.0** | _31.6_ | _18.9_ | 69.5 |

Table 10: Performance (%) on LoCoMo (Part 2/2). The adversarial number only represents refusal performance partially (see §6.2 for details).

| Methods | Open-Domain (841) | | | Temporal (96) | | | Adversarial (446) | | |
|---|---|---|---|---|---|---|---|---|---|
| | F1 | BLEU-1 | LLM-J | F1 | BLEU-1 | LLM-J | F1 | BLEU-1 | LLM-J |
| Oracle Message | 43.8 | 30.5 | 86.0 | 36.7 | 23.7 | 59.4 | 70.6 | 70.7 | 70.6 |
| Full-Context | 36.0 | 23.9 | 88.3 | 25.6 | 15.9 | 54.2 | 52.2 | 52.1 | 55.0 |
| *Large Embedding Models* | | | | | | | | | |
| Qwen3-Embed-8B | 27.9 | 19.6 | 67.3 | 25.1 | 15.3 | 45.8 | 65.9 | 66.0 | 66.8 |
| NV-Embed-v2 (7B) | 36.8 | 24.6 | 82.6 | 26.2 | 16.2 | 51.0 | 60.8 | 60.8 | 61.2 |
| w/ Top-3 Messages | 33.7 | 23.3 | 71.7 | 23.1 | 13.1 | 43.8 | 72.9 | 72.9 | 73.1 |
| *Structure-Augmented Memory* | | | | | | | | | |
| Mem0 | 38.2 | **27.5** | 57.2 | 24.1 | 15.7 | 46.9 | 8.3 | 10.4 | 46.6 |
| Graphiti | 21.9 | 15.6 | 45.0 | 23.0 | 14.6 | 45.8 | **83.2** | **83.4** | **83.0** |
| HippoRAG 2 | 35.5 | 23.9 | 81.7 | **28.8** | **18.5** | 56.3 | 62.6 | 62.6 | 65.7 |
| *Ours* | | | | | | | | | |
| REMem-I | **39.1** | 26.4 | 83.5 | 25.5 | 17.5 | 56.3 | 61.7 | 62.0 | 66.8 |
| REMem-S | 37.3 | 24.6 | **85.1** | 28.2 | **18.5** | **63.5** | 62.1 | 61.8 | 65.3 |

## C.2 REALTALK

Overall, REALTALK proves to be more challenging than LoCoMo, as evidenced by the consistently lower performance of all methods. Constructed from real human interactions, it naturally contains more noise and casual expressions. The overall performance distribution is similar to that observed on LoCoMo. NV-Embed-v2 remains a strong baseline as an embedding model, while structure-enhanced memory methods fall short in comparison. REMem is the only one that surpasses the Full-Context method, particularly on tasks involving temporal reasoning.

Table 11: Performance (%) on REALTALK.

| Methods | Overall (728) | | | Multi-hop (301) | | | Commonsense (108) | | | Temporal (319) | | |
|---|---|---|---|---|---|---|---|---|---|---|---|---|
| | F1 | BLEU-1 | LLM-J | F1 | BLEU-1 | LLM-J | F1 | BLEU-1 | LLM-J | F1 | BLEU-1 | LLM-J |
| Full-Context | 25.3 | 18.6 | 65.1 | 27.8 | 21.7 | 60.1 | 19.5 | 15.1 | 55.6 | 24.9 | 16.8 | 73.0 |
| *Large Embedding Models* | | | | | | | | | | | | |
| Qwen3-Embed-8B | 20.2 | 14.9 | 52.5 | 19.6 | 16.2 | 41.5 | 14.5 | 11.8 | 39.8 | 22.7 | 14.8 | 67.1 |
| NV-Embed-v2 (7B) | 23.8 | 17.7 | 59.5 | 24.5 | 19.4 | 51.2 | 16.6 | 13.0 | 48.2 | 25.6 | **17.7** | 71.2 |
| *Structure-Augmented Memory* | | | | | | | | | | | | |
| Mem0 | 9.8 | 7.2 | 14.3 | 14.0 | 9.6 | 20.9 | 12.0 | 9.0 | 22.2 | 5.1 | 4.3 | 5.3 |
| Graphiti | 15.1 | 11.5 | 35.3 | 19.6 | 15.5 | 39.5 | 16.7 | 13.2 | 44.4 | 10.5 | 7.1 | 28.2 |
| HippoRAG 2 | 21.9 | 16.2 | 55.8 | 26.1 | 20.9 | 51.8 | **18.2** | 13.7 | **54.6** | 19.2 | 12.6 | 59.9 |
| *Ours* | | | | | | | | | | | | |
| REMem-I | 25.6 | 18.1 | 63.7 | 26.6 | 21.6 | 55.8 | 16.1 | 13.0 | 45.4 | **27.9** | 16.5 | **77.4** |
| REMem-S | **26.2** | **19.2** | **65.3** | **28.6** | **23.1** | **58.8** | **18.2** | **14.7** | 53.7 | 26.7 | 17.1 | 75.2 |

## C.3 COMPLEX-TR

The performance on Complex-TR is shown in Table 12. On Complex-TR, embedding models remain strong baselines. Among structure-augmented memory methods, HippoRAG 2 slightly outperforms the embedding model (J +1.1), likely due to its alignment with the entity-centric nature of this dataset. Compared to REMem, the only competitive method is NV-Embed-v2 w/ TISER, which employs multi-step temporal reasoning prompts and even outperforms REMem-I on the time-to-event task (J +1.4), but falls short on average (J −1.3). Compared with the event-to-event type, the time-to-event type requires more direct temporal resolution, which better suits TISER.

REMem primarily focuses on information extraction and agentic retrieval. For the reasoning component, we employ straightforward prompts (Appendix D) that leverage the context to answer questions. When we adopt TISER as the reasoning prompt for the final step, REMem w/ TISER achieves

further improvements and delivers the highest performance (F1 +7.3, J +2.4, compared to REMem) overall. Additionally, for such complex reasoning tasks, REMem-S lacks multi-hop reasoning capabilities (J −6.0), which demonstrates the necessity of employing agentic retrieval for autonomous reasoning.

Table 12: Performance (%) on Complex-TR.

| Methods | Avg | | | Time to Event (543) | | | Event to Event (457) | | |
|---|---|---|---|---|---|---|---|---|---|
| | F1 | BLEU-1 | LLM-J | F1 | BLEU-1 | LLM-J | F1 | BLEU-1 | LLM-J |
| Full-context | 74.2 | 68.0 | 81.6 | 73.0 | 65.6 | 80.3 | 75.6 | 70.8 | 83.2 |
| *Large Embedding Models* | | | | | | | | | |
| Qwen3-Embed-8B | 77.1 | 71.4 | 80.9 | 74.5 | 68.8 | 79.0 | 80.2 | 74.5 | 83.2 |
| NV-Embed-v2 (7B) | 77.5 | 71.9 | 80.4 | 77.0 | 70.8 | 79.2 | 78.1 | 73.2 | 81.8 |
| w/ TISER | 88.1 | 83.6 | 88.3 | 88.9 | 83.5 | 90.4 | 87.1 | 83.7 | 85.8 |
| *Structure-Augmented Memory* | | | | | | | | | |
| Mem0 | 43.1 | 35.1 | 41.0 | 43.5 | 35.1 | 40.9 | 42.6 | 35.1 | 41.1 |
| Graphiti | 76.6 | 71.4 | 78.8 | 77.3 | 71.1 | 80.3 | 75.6 | 71.7 | 77.0 |
| HippoRAG 2 | 78.2 | 72.7 | 81.5 | 78.0 | 71.4 | 81.8 | 78.5 | 74.4 | 81.2 |
| *Ours* | | | | | | | | | |
| REMem-I | 83.3 | 77.6 | 89.6 | 80.3 | 73.2 | 89.0 | 86.9 | 82.8 | 90.4 |
| w/ TISER | 90.6 | 86.0 | 92.0 | 92.5 | 87.1 | 94.3 | 88.1 | 83.8 | 89.3 |
| REMem-S | 78.5 | 72.7 | 82.6 | 78.6 | 71.6 | 84.2 | 78.5 | 74.0 | 80.7 |

## C.4 TEST OF TIME (SEMANTIC)

The performance on the Test of Time is shown in Table 13. Here, Full-Context directly uses the original prompt from the dataset, which contains hundreds of facts per question on average. Since Test of Time employs anonymous entity and relation labels, we also use BM25 as the retriever in addition to the embedding model. The results show that even the improved GPT-5-chat still falls short as a Full-Context method when compared with its predecessor, the economical GPT-4.1-mini (EM +4.8). HippoRAG 2 heavily relies on graph traversal, and due to its lack of understanding of complex logic, its performance falls short of even embedding models. TISER continues to serve as a strong baseline, while REMem is the only method that surpasses 90%, achieving an +8.2 EM improvement over Full-Context with TISER. Beyond Table 13, we further observe that REMem-I, when paired with stronger GPT-5-chat, can address nearly all eight types of challenges in this benchmark (EM 99.0), underscoring the rationality of the agentic retrieval design.

## C.5 SEMANTIC MEMORY CAPABILITY

Although episodic memory is a crucial aspect of long-term memory and the main focus of this paper, semantic memory should not be overlooked as another aspect. We select MuSiQue and 2Wiki from the benchmarks used in HippoRAG 2 to evaluate this capability. From Table 7, we observe that Mem0 performs substantially worse than both NV-Embed-v2 and REMem across all metrics, with results nearly collapsing. This performance highlights limitations in Mem0's memory construction process: a major part of the extracted facts is not added to its memory, and its use of a fixed dialog extraction pipeline is poorly aligned with the structure of passages that incorporate world knowledge. Mem0 frequently extracts incomplete or decontextualized sentences in which even the subject is ambiguous, e.g.,. "*Managed La Liga clubs including Barcelona, Atlético Bilbao, Atlético Madrid, and Real Zaragoza*" is extracted from a paragraph titled "*Ferdinand Daučík*", and then this extracted sentence is not added or updated to the memory. For comparison, though our method primarily focuses on episodic memory and reasoning, it still achieves performance comparable to a strong embedding model on this task, demonstrating its strong extensibility.

Table 13: Exact match (%) on Test of Time (semantic). GPT-4.1-mini is the default LLM. While Full-Context adopts the full prompt from the original dataset, BM25 and NV-Embed-v2 provide shortened prompts by retrieving the top-$k$ facts. BA: Before-after. ET: Event at time $t$. EW: Event at what time. FL: First-last. EA: Event at the time of another event. NE: Number of events in time interval. RD: Relation duration. TL: Timeline. Each category contains 350 samples. * denotes using a 160-sample subset as an approximation due to the high cost of full-dataset evaluation.

| Methods | Avg | BA | ET | EW | FL | EA | NE | RD | TL |
|---|---|---|---|---|---|---|---|---|---|
| Full-Context (GPT-4.1-mini) | 79.7 | 72.0 | 86.9 | 98.0 | 81.1 | 83.7 | 76.3 | 93.4 | 46.3 |
| w/ TISER | 84.9 | 88.3 | 91.4 | 97.7 | 89.1 | 90.6 | 64.0 | 94.6 | 63.4 |
| Full-Context (GPT-5-chat) | 84.5 | 80.0 | 90.0 | 99.4 | 84.9 | 91.1 | 90.0 | 82.3 | 58.0 |
| BM25 | 80.8 | 71.1 | 77.7 | 98.6 | 92.3 | 64.3 | 65.1 | 98.3 | 79.1 |
| Qwen-Embed-8B | 70.3 | 53.1 | 88.3 | **100.0** | 72.6 | 42.9 | 71.7 | 98.0 | 35.7 |
| NV-Embed-v2 (7B) | 68.9 | 43.1 | 86.6 | **100.0** | 74.6 | 48.0 | **75.7** | 96.9 | 26.6 |
| w/ TISER | 68.9 | 48.0 | 86.0 | **100.0** | 74.6 | 51.7 | 67.4 | 96.9 | 26.3 |
| HippoRAG 2* | 66.9 | 60.0 | 90.0 | **100.0** | 75.0 | 45.0 | 50.0 | **100.0** | 15.0 |
| REMem-I | **93.1** | 97.4 | **99.4** | **100.0** | 92.0 | **97.4** | 71.4 | 94.0 | **92.9** |
| w/ TISER | 90.6 | **98.6** | **99.4** | 98.6 | **95.4** | 93.7 | 68.3 | 85.1 | 85.4 |
| REMem-S | 72.5 | 74.6 | 78.0 | **100.0** | 90.9 | 64.3 | 52.0 | 99.7 | 20.9 |

Table 14: Performance (%) on QA tasks with semantic memory.

| | Avg | | | MuSiQue $(1,000)$ | | | 2Wiki $(1,000)$ | | |
|---|---|---|---|---|---|---|---|---|---|
| Methods | F1 | BLEU-1 | LLM-J | F1 | BLEU-1 | LLM-J | F1 | BLEU-1 | LLM-J |
| NV-Embed-v2 | 38.2 | 30.5 | **57.5** | 37.5 | 31.9 | **56.3** | 38.8 | 29.1 | **58.6** |
| Mem0 | 6.9 | 5.1 | 8.0 | 7.4 | 5.7 | 9.2 | 6.3 | 4.5 | 6.8 |
| REMem | **38.2** | **32.1** | 55.3 | **37.9** | **33.5** | 53.2 | **38.6** | 30.8 | 57.4 |

## C.6 ABLATION STUDY

More detailed results of the ablation study are presented in Table 15 and Table 16. Overall, gists provide the primary context (J $-27.3\%$ on LoCoMo and $-8.7\%$ on Complex-TR), while facts offer indispensable supplementary support. In particular, for LoCoMo, REMem-I w/o Facts achieved comparable performance to REMem across most tasks, but showed a clear drop in multi-hop questions (J $-7.1\%$). This highlights the crucial role of phrase nodes in bridging concepts across sessions and facilitating effective graph exploration.

Table 15: Ablation study on LoCoMo.

| | Avg | | Single-Hop | | Multi-Hop | | Open-Domain | | Temporal | | Adversarial | |
|---|---|---|---|---|---|---|---|---|---|---|---|---|
| Methods | F1 | LLM-J | F1 | LLM-J | F1 | LLM-J | F1 | LLM-J | F1 | LLM-J | F1 | LLM-J |
| REMem-I | **42.4** | 76.2 | 37.9 | 81.3 | **32.6** | **70.2** | 39.1 | 83.5 | 25.5 | 56.3 | 61.7 | 66.8 |
| w/o Gists | 31.7 | 48.9 | 19.7 | 69.8 | 20.5 | 42.2 | 12.0 | 24.9 | 14.5 | 28.1 | **88.3** | **88.1** |
| w/o Facts | 42.0 | 74.1 | **45.7** | 81.9 | 28.2 | 63.1 | **41.7** | 84.0 | 25.6 | 54.2 | 52.2 | 61.0 |
| REMem-S | 41.3 | **77.5** | 35.4 | **86.0** | 31.6 | 69.5 | 37.3 | **85.1** | **28.2** | **63.5** | 62.1 | 65.3 |

## C.7 PERFORMANCE BY TEMPORAL CATEGORY

We present performance metrics by temporal category to provide more analysis for temporal and non-temporal questions. We set a few temporal categories, as shown in Table 17, and instruct GPT-4.1-mini to classify each query into one of the categories.

Then, we report REMem-I's performance on LoCoMo in Table 18 according to the above categories, where 'temporal' is the average value of all temporal categories, and 'overall' is the average value

Table 16: Ablation study on Complex-TR.

| Methods | Avg | | | Time to Event (543) | | | Event to Event (457) | | |
|---|---|---|---|---|---|---|---|---|---|
| | F1 | BLEU-1 | LLM-J | F1 | BLEU-1 | LLM-J | F1 | BLEU-1 | LLM-J |
| REMem-I | **83.3** | **77.6** | **89.6** | **80.3** | **73.2** | **89.0** | 86.9 | 82.8 | **90.4** |
| w/o Gists | 80.3 | 75.9 | 80.9 | 73.7 | 69.1 | 75.3 | **88.1** | **84.0** | 87.5 |
| w/o Facts | 80.5 | 74.5 | 87.2 | 79.3 | 72.7 | 86.6 | 81.9 | 76.6 | 88.0 |
| REMem-S | 78.5 | 72.7 | 82.6 | 78.6 | 71.6 | 84.2 | 78.5 | 74.0 | 80.7 |

Table 17: Our defined temporal categories on LoCoMo.

| Category | Count | Description | Query Example |
|---|---|---|---|
| Existence Check | 14 | Check for the presence of temporal facts, given a specific time frame | Did Andrew have a pet dog during March 2023? |
| Event Timing | 298 | Determine the specific time points or intervals when an event occurred | When did Melanie paint a sunrise? |
| Event Attributes | 302 | Identify attributes or characteristics of an event, e.g., location, participants, etc. | What are some changes Caroline has faced during her transition journey? |
| Order | 22 | Understand temporal relations between events, e.g., sequence, concurrency, overlap | What did Melanie do after the road trip to relax? |
| Duration | 44 | Determine the duration of an event or the time interval between two events | How long has Caroline had her current group of friends for? |
| Aggregation | 30 | Count occurrences of an event within a time frame, or count specific attributes of events | How many times has Melanie gone to the beach in 2023? |
| Other | 6 | Other temporal reasoning tasks not covered by the above categories | Would Caroline want to move back to her home country soon? |
| Non-temporal | 1270 | No temporal reasoning is required, but other situational elements | Where did Oliver hide his bone once? |

of 'temporal' and 'none' categories. These results suggest that the performance of REMem-I on non-temporal questions is on par with the temporal ones.

# D   PROMPTS

This section shows details of REMem. The prompts used for gist and fact extraction are illustrated in Figure 3 and Figure 4. The prompts for tool selection, along with the corresponding tool descriptions, are presented in Figures 5–8. The tool selector selects one of the available tools at each step based on their descriptions.

# E   IMPLEMENTATION DETAILS

We replicated using open-source versions of Mem0 (Chhikara et al., 2025) and Graphiti (Rasmussen et al., 2025) rather than proprietary ones, aligning settings as closely as feasible, including the backbone LLM, embedding model, and the scale of contexts. Since we observed Mem0 frequently autonomously choosing to reject adding input text to memory, we opted for finer granularity: adding memory at the message level to encourage more information to be incorporated into memory. For Graphiti, we found that frequent LLM and embedding calls incurred excessive time and economic costs, and added memory at the session level. Subsequently, Graphiti indexes information within the session into multiple facts for subsequent retrieval. Thus, both Mem0 and Graphiti store information at the fact/sentence level rather than the session level.

Table 18: The performance on LoCoMo by temporal categories.

| Temporal category | # of samples | F1 | BLEU-1 | LLM-J |
|---|---|---|---|---|
| Overall | 1986 | 42.4 | 32.7 | 76.2 |
| Non-temporal | 1086 | 41.8 | 32.4 | 75.0 |
| Temporal | 900 | 43.2 | 33.1 | 77.7 |
| Existence Check | 15 | 59.5 | 11.9 | 86.7 |
| Event Attributes | 450 | 45.9 | 37.9 | 76.7 |
| Event Timing | 298 | 39.1 | 26.0 | 82.6 |
| Aggregation | 35 | 39.5 | 33.9 | 60.0 |
| Duration | 50 | 35.2 | 26.8 | 72.0 |
| Order | 49 | 48.4 | 45.0 | 71.4 |
| Other | 3 | 42.4 | 37.8 | 100.0 |

Table 19: The running time and memory usage.

| REMem-I | Index Time (s) | Inference Time / Query (s) | Max Memory Usage (GB) |
|---|---|---|---|
| LoCoMo | 3,604.2 | 4.3 | 2.6 |
| Complex-TR | 378.7 | 12.6 | 1.3 |

# F ANALYSIS

## F.1 COMPARATIVE ANALYSIS: REMEM VS. TISER

We compare REMem with NV-Embed-v2 w/ TISER (Bazaga et al., 2025) on Complex-TR and show a few examples. NV-Embed-v2 w/ TISER is denoted as TISER in this paragraph. REMem outperformed TISER typically by handling multi-hop temporal reasoning more comprehensively and recovering the full set of required entities. For example, when asked "*Where was Nancy L. Ross educated before ASU?*", the gold answer was "*BC Cancer Research Centre*", and ours produced "*Virginia Tech*" and "*BC Cancer Research Centre*", which the judge accepted as correct, while TISER only returned Virginia Tech. Conversely, TISER surpassed REMem, usually by pinpointing the exact target within the specified time window, whereas ours produced over-verbose lists or drifted to the wrong temporal hop.

For instance, in the query "*Where was Barack Obama educated after State Elementary School Menteng 01?*", the gold was "*Punahou School*". TISER returned exactly that, while REMem instead listed later universities. This issue is most likely due to the ambiguity of the terms "*before*" and "*after*", which may introduce uncertainty in determining whether an inequality should include equality when using the provided tools in REMem, and the resulting tool calls do not always align with the intended meaning of the question. However, the precise relationship between "*before*" and "*after*" can often be more reliably inferred from the retrieved context, since it can be directly inferred within the temporal scope of a sequence of events.

## F.2 TIME AND SPACE EFFICIENCY

In the indexing phase, Graphiti is two orders of magnitude less efficient than Mem0 and REMem. This may be because its processing of each episode involves multiple rounds of calls to both the generative LLM and the embedding model. In the inference phase, the single-step runtime of REMem is comparable to Mem0 and Graphiti, while its multi-step runtime grows linearly with the number of steps.

We further report the runtime and memory usage of REMem in Table 19, where it reports the memory consumption of REMem itself. If an LLM or embedding model service is deployed locally, its main memory and GPU memory usage are counted separately. Experiments were conducted on a server with dual AMD EPYC 7643 48-Core CPUs (96 hardware threads), 4× NVIDIA A100-SXM4-80GB GPUs, and 1 TB of system memory. The inference time for each query is affected by many factors, including the number of workers and the response time of the LLM service. The reported

Table 20: Token usage on LoCoMo ($1,986$ queries) using OpenAI o200k_base encoding. The estimated cost is calculated using the standard GPT-4.1-mini pricing.

| | Indexing Phase | | Inference Phase | | Total | Est. Cost (USD) |
|---|---|---|---|---|---|---|
| Method | Input | Output | Input | Output | | |
| NV-Embed-v2 | N/A | N/A | 1.19M | 0.17M | 1.36M | $ 0.75 |
| REMem-I | 0.90M | 0.72M | 18.10M | 0.79M | 20.51M | $10.02 |
| REMem-S | 0.90M | 0.72M | 1.83M | 0.18M | 3.63M | $ 2.53 |

time excludes any LLM or embedding caching and does not use multithreading. With caching or multithreading enabled, the actual throughput would be higher.

### F.3 TOKEN USAGE

The token usage of REMem on LoCoMo is shown in Table 20. REMem-S exhibits token consumption nearly equivalent to the embedding baseline during the inference phase, while the token consumption of REMem-I increases with the number of iterations.

### F.4 EXAMPLE OF THE EXTRACTION

We present a passage from MuSiQue, along with the extraction results from Mem0 and REMem, in Table 21. In comparison, Mem0's extraction results exhibit a lack of factual coverage, which may hinder the subsequent comprehension and QA task. The extraction from REMem is more comprehensive, especially as the last (14th) fact, even inferring that the season close of RFEF (2002-2003) is June 2003. Mem0 stores memories prefixed with "user", assuming the working scenario is always conversational messages, yet this is unnecessary for long passages that describe objective knowledge.

Table 21: A MuSiQue passage and its extraction results from Mem0 and REMem.

| **Passage: Lionel Messi** |
|---|
| After a year at Barcelona's youth academy, La Masia, Messi was finally enrolled in the Royal Spanish Football Federation (RFEF) in February 2002. Now playing in all competitions, he befriended his teammates, among whom were Cesc Fàbregas and Gerard Piqué. After completing his growth hormone treatment aged 14, Messi became an integral part of the 'Baby Dream Team", Barcelona's greatest-ever youth side. During his first full season (2002–03), he was top scorer with 36 goals in 30 games for the Cadetes A, who won an unprecedented treble of the league and both the Spanish and Catalan cups. The Copa Catalunya final, a 4–1 victory over Espanyol, became known in club lore as the partido de la máscara, the final of the mask. A week after suffering a broken cheekbone during a league match, Messi was allowed to start the game on the condition that he wear a plastic protector; soon hindered by the mask, he took it off and scored two goals in 10 minutes before his substitution. At the close of the season, he received an offer to join Arsenal, his first from a foreign club, but while Fàbregas and Piqué soon left for England, he chose to remain in Barcelona. |
| **Mem0 Extraction** |
| User mentioned Lionel Messi's early football career details including his time at Barcelona's youth academy La Masia, enrollment in the Royal Spanish Football Federation in February 2002, friendship with teammates Cesc Fàbregas and Gerard Piqué, completion of growth hormone treatment at age 14, being top scorer with 36 goals in 30 games during the 2002-03 season for Cadetes A, winning a treble of league and Spanish and Catalan cups, playing with a plastic mask after a broken cheekbone and scoring two goals, and receiving an offer from Arsenal but choosing to stay in Barcelona. |
| **REMem Gists** |

Table 21: A MuSiQue passage and its extraction results from Mem0 and REMem. (Continued)

1. Lionel Messi was enrolled in the Royal Spanish Football Federation (RFEF) in February 2002 after a year at Barcelona's youth academy, La Masia.

2. Messi befriended his teammates Cesc Fàbregas and Gerard Piqué while playing in all competitions.

3. After completing his growth hormone treatment at age 14, Messi became an integral part of Barcelona's greatest-ever youth side, the 'Baby Dream Team'.

4. During his first full season (2002–03), Messi was the top scorer with 36 goals in 30 games for the Cadetes A.

5. The Cadetes A won an unprecedented treble of the league and both the Spanish and Catalan cups during the 2002–03 season.

6. The Copa Catalunya final, a 4–1 victory over Espanyol, became known as the partido de la máscara, the final of the mask.

7. A week after suffering a broken cheekbone during a league match, Messi was allowed to start the Copa Catalunya final on the condition that he wear a plastic protector.

8. Messi took off the plastic protector and scored two goals in 10 minutes before his substitution in the Copa Catalunya final.

9. At the close of the 2002–03 season, Messi received an offer to join Arsenal, his first from a foreign club.

10. Messi chose to remain in Barcelona while Fàbregas and Piqué soon left for England.

**REMem Facts**

1. (Lionel Messi, was enrolled in, the Royal Spanish Football Federation (RFEF)) [point in time: 2002-02]

2. (Lionel Messi, befriended, Cesc Fàbregas and Gerard Piqué)

3. (Lionel Messi, became an integral part of, Barcelona's greatest-ever youth side, the 'Baby Dream Team') [point in time: 2002-03]

4. (Lionel Messi, was the top scorer with, 36 goals in 30 games for the Cadetes A) [point in time: 2002-03]

5. (Cadetes A, won, an unprecedented treble of the league and both the Spanish and Catalan cups) [point in time: 2002-03]

6. (Copa Catalunya final, was a victory over, Espanyol) [point in time: 2002-03]

7. (Copa Catalunya final, became known as, the partido de la máscara, the final of the mask)

8. (Lionel Messi, suffered, a broken cheekbone during a league match) [point in time: 2002-03]

9. (Lionel Messi, was allowed to start, the Copa Catalunya final on the condition that he wear a plastic protector) [point in time: 2002-03]

10. (Lionel Messi, took off, the plastic protector) [point in time: 2002-03]

11. (Lionel Messi, scored, two goals in 10 minutes before his substitution in the Copa Catalunya final) [point in time: 2002-03]

12. (Lionel Messi, received an offer to join, Arsenal, his first from a foreign club) [point in time: 2003-06]

13. (Lionel Messi, chose to remain in, Barcelona)

14. (Cesc Fàbregas and Gerard Piqué, left for, England) [point in time: 2003-06]

F.5  GRAPH PROPERTY

For the evaluated four benchmarks, Table 22 reports the scale of the constructed memory graphs, including the numbers of phrase and gist nodes, edges, triples, and associated token counts. The statistics show that LoCoMo, REALTALK, and Complex-TR induce large, densely connected graphs with many context and synonymy edges. For Test of Time, the graph is much smaller and contains no gist-level annotations, since its statements involve anonymous entities and are already highly formalized, so we extract only fact-level information.

Table 22: The graph properties on evaluated benchmarks: average values for each graph.

| | LoCoMo | REALTALK | Complex-TR | Test of Time |
|---|---|---|---|---|
| # of phrase nodes | 777.5 | 974.0 | 1,066.0 | 16.0 |
| # of gist nodes | 730.1 | 889.0 | 1,095.0 | – |
| # of relation edges | 736.2 | 891.0 | 1,062.0 | – |
| # of context edges | 25,172.4 | 56,332.0 | 2,190.0 | – |
| # of synonymy edges | 1,082.2 | 606.0 | 748.0 | – |
| # of triples | 763.9 | 917.0 | 1,095.0 | 275.4 |
| # of input tokens | 15,965.8 | 19,250.0 | 26,158.0 | 4,109.7 |
| # of phrase-node tokens | 5,105.8 | 8,350.0 | 5,536.0 | 32.1 |
| # of gist-node tokens | 21,743.8 | 30,106.0 | 26,514.0 | – |
| Node degree of phrase nodes | 34.0 | 59.7 | 4.1 | 9.7 |
| Node degree of gist nodes | 34.4 | 63.4 | 2.0 | – |

## Gist Extraction Instruction & Demonstration

You are a meticulous information extractor. Your purpose is to distill personal episodic memories from messages into a structured JSON format.

## Core Task
For the given message(s), identify every individual fact, event, or claim. Restate each one as a concise, self-contained English sentence.

## Input Format
The user will provide the current time and the message text. You MUST use the `current_time` to resolve any relative temporal expressions
(e.g., "yesterday", "last week").

## Output Format
- Your output MUST be a single, valid JSON object.
- The JSON object must contain one key: `"gists"`.
- The value of `"gists"` is a list of strings.
- Do not add any explanations, comments, or trailing commas.

### Rules for Gists
1.  **Decomposition:** Decompose complex sentences into multiple gists. Each gist should represent a single atomic fact or event.
2.  **Timestamp Prefix:** Begin every gist with the message's timestamp in square brackets, e.g., `[20 January 2025, 2:28 pm]`.
3.  **Temporal Resolution:** After any temporal reference, add the fully-resolved absolute date or date range in parentheses.
    -   *Time Point Example*: `...last Thursday (16 January 2025).`
    -   *Duration Example*: `...last week (12 January 2025 to 18 January 2025).`
4.  **Completeness:** Capture ALL details for each fact: participants, actions, objects, quantities, locations, intentions, etc.
5. Infer reasonable details about the above dimensions as many as possible for later retrieval, but do NOT invent new information.

---

Input 1:
```
Date: 3:57 pm on 20 January, 2024
Alice: I fixed the fence last Monday, then bought 3 cows from Peter on Jan 15th
```
Output 1:
```json
{
  "gists": [
    "[20 January 2024, 3:57 pm] Alice fixed the fence last Monday (15 January 2024).",
    "[20 January 2024, 3:57 pm] Alice bought 3 cows from Peter on Jan 15th (15 January 2024)."
  ]
}
```

Input 2:
```
Date: 2:28 pm on 20 January, 2025
Bob: I met with my advisor last Thursday morning and submitted the proposal two days later.
```
Output 2:
```json
{
  "gists": [
    "[20 January 2025, 2:28 pm] Bob met with his advisor last Thursday morning (16 January 2025).",
    "[20 January 2025, 2:28 pm] Bob submitted the proposal two days later (18 January 2025)."
  ]
}
```

Figure 3: The prompts for gist extraction. The instructions and demonstrations are marked in different colors.

## Fact Extraction Instruction & Demonstration

```
episodic_system = """Extract structured facts from personal episodic memory messages.

Return JSON with the unique key `facts` - a list where each fact has:
- `subject` (str): entity performing/experiencing the action
- `predicate` (str) : the action, relationship, or state
- `object` (str): entity/concept being acted upon
- `qualifiers` (dict): format in `%d %B %Y, %I:%M %p` is preferred for each of the following properties
  - `record_time`: (str, required) when message was created
  - `point_in_time`: (str, optional) only used to indicate a point in time when event occurred; if this is used, ignore
`start_time` and `end_time`
  - `start_time`: (str, optional) event start, used to indicate a time range
  - `end_time`: (str, optional) event end, used to indicate a time range

Use short "event handles" (e.g. "fence fixing") for reusability. Connect events through shared entities.

# Rules
- Capture all factual claims, quantities, temporal references, and relationships
- Include everything; err on the side of inclusion
- Use text-supported interpretations, not assumptions, to avoid hallucinations
- Leverage additional gists when they are provided
- Always include `record_time` from provided date/time
- Return valid JSON only, no extra keys or comments
```

```
Input:
```
Date: 3:57 pm on 20 Jan, 2024
Alice: I fixed the fence last Sunday, then bought 3 cows from Peter on Jan 15th
```

Output:
```json
{
  "facts": [
    {
      "subject": "Alice",
      "predicate": "completed task",
      "object": "fence fixing",
      "qualifiers": {
        "record_time": "20 Jan 2024, 3:57 pm",
        "point_in_time": "14 Jan 2024"
      }
    },
    {
      "subject": "Alice",
      "predicate": "purchased",
      "object": "3 cows",
      "qualifiers": {
        "record_time": "20 Jan 2024, 3:57 pm",
        "point_in_time": "15 Jan 2024"
      }
    },
    {
      "subject": "cow purchase",
      "predicate": "source",
      "object": "Peter",
      "qualifiers": {
        "record_time": "20 Jan 2024, 3:57 pm",
        "point_in_time": "15 Jan 2024"
      }
    }
  ]
}
```
```

Figure 4: The prompts for fact extraction. The instructions and demonstrations are marked in different colors.

## Tool Selection: Instruction & Demonstration

You are an intelligent agent exploring a large knowledge graph (KG) to answer a query. Select the best tool and its parameters given the current contexts.

**Instructions:**
Understand the contexts for previous and current steps and consider whether we need to explore on the KG or output the answer.
Pay attention to the observations from previous steps - they tell you what was found and how much information was retrieved.
Plan ahead and select one best tool from the available tools and provide its parameters for current step.

Tool Use Strategy:
1. Initial Retrieval: Begin by using `semantic_retrieve` or `lexical_retrieve` to understand the data schema and gather preliminary context. The goal is to identify the structure of the KG. For complex queries, break them down into simpler sub-queries to guide the next steps.
- Prioritize `semantic_retrieve` over `lexical_retrieve`, unless the query specifically targets entity identifiers or requires an exact match.
- Note: These tools return truncated content. This initial scan is for identifying key information (e.g., entity IDs, time ranges) required for a more detailed search.
2. Focused Exploration: Next, use `find_gist_contexts` or `find_entity_contexts` to retrieve specific, detailed information.
- Populate the parameters of these tools (e.g., subject, object, limit, ordering) using the key information discovered in the initial retrieval step.
- Prioritize `find_gist_contexts` over `find_entity_contexts`. Use the latter one ONLY when a query specifically targets entities with a well-defined KG schema you've seen; otherwise, DO NOT try to match the exact phrase using `find_entity_contexts`.
3. Answer Generation: Once you have gathered information and are confident in the answer, use the `output_answer` tool to provide the final result.

Temporal Reasoning Example:
- Find facts start after 1950: start_time='1950', start_operator='gt'
- Find facts end before 1960: end_time='1960', end_operator='lt'
- Find facts exist in specific point '1955': start_time='1955', start_operator='le', end_time='1955', end_operator='ge'
- Find facts only exist within [1950, 1960]: start_time='1950', start_operator='ge', end_time='1960', end_operator='le'
- Find facts that overlap with [1950, 1960]: start_time='1960', start_operator='le', end_time='1950', end_operator='ge'
(note: a special case where the start time is larger than the end time)

**Output Format:**
Respond with a JSON object, e.g.,
{{
    "reasoning": "Brief explanation of why this tool and parameters for this step",
    "function": "exact_tool_name_from_list",
    "parameters": {{
        "name": "value",
    }}
}}

For the query 'Find the entity that was the R11 of E57 immediately after E95 was the R11 of E57.', each step are like this:
Step 1: {{"reasoning": "Find the end time of the event `E95 was the R11 of E57`, which is the start time of the next event that `was the R11 of E57`", "function": "lexical_retrieve", "parameters": {{"query": "E95 was the R11 of E57"}}}}
  - Observation: Retrieved 5 gists and 3 triples via BM25 lexical matching, showing 5 gists and 3 triples
  - Truncated top contents:
    1. E95 was appointed as the R11 of E57 on January 15, 2020. This position involves managing the technical operations and strategic planning for entity E57.
    2. (E95, was the R11 of, E57) {{"start_time": "2020-01-15", "end_time": "2022-03-20"}}
    3. The R11 role for E57 was held by E95 from 2020 to 2022, during which time significant improvements were made to the organization's infrastructure.

Step 2: {{"reasoning": "Find the next event that was the R11 of E57 after the retrieved time t1", "function": "find_entity_contexts", "parameters": {{"object": "E57", "relation": "was the R11 of", "start_time": "t1"}}}}
  - Observation: Found 12 connected gists and 8 connected triples for entity E57 (filtered by time: t1 to any), showing 15 gists and 15 triples
  - Truncated top contents:
    1. (E0, was the R11 of, E57) {{"start_time": "2022-03-21", "end_time": "2023-06-15"}}
    2. E0 succeeded E95 as the R11 of E57 on March 21, 2022, immediately after E95's term ended. E0 brought new perspectives and strategies to the role.
    3. (E95, was the R11 of, E57) {{"start_time": "2020-01-15", "end_time": "2022-03-20"}}

Step 3: {{"reasoning": "The entity that is closest to t1 and matches `was the R11 of E57` is E0", "function": "output_answer", "parameters": {{"answer": "E0"}}}}

Figure 5: The prompts for tool selection. The instructions and demonstrations are marked in different colors.

```
{
    "type": "function",
    "name": "output_answer",
    "description": "When you believe you have information, analyze retrieved information and corresponding questions
meticulously, answer the original query and end the search process.",
    "parameters": {
        "type": "object",
        "properties": {
            "answer": {
                "type": "string",
                "description": "Present a concise, definitive response, devoid of additional elaborations. Can be string, integer,
or any format required by the original query."
            }
        },
        "required": ["answer"]
}
```

```
{
    "type": "function",
    "name": "semantic_retrieve",
    "description": "Search for the most semantically relevant 'gists' and 'facts' (triples) in the KG using embedding
similarity. It returns the top results from both types at once. Use when you need semantic understanding and conceptual
matching.",
    "parameters": {
        "type": "object",
        "properties": {
            "query": {
                "type": "string",
                "description": "Search query string"
            },
            "start_time": {
                "type": "string",
                "description": "The start of a time range to filter facts by their temporal qualifiers"
            },
            "end_time": {
                "type": "string",
                "description": "The end of a time range to filter facts by their temporal qualifiers"
            },
            "start_operator": {
                "type": "string",
                "description": "Operator for start time comparison (lt, le, ge, gt, eq)."
            },
            "end_operator": {
                "type": "string",
                "description": "Operator for end time comparison (lt, le, ge, gt, eq)."
            }
        },
        "required": ["query"]
}
```

Figure 6: The tool descriptions for *output_answer* and *semantic_retrieve*.

```
{
  "type": "function",
  "name": "lexical_retrieve",
  "description": "Search for the most relevant 'gists' and 'facts' (triples) in the KG based on BM25 scoring. It returns the
top results from both types at once. Use when you need keyword-based or exact term matching, e.g., identifiers.",
  "parameters": {
    "type": "object",
    "properties": {
      "query": {
        "type": "string",
        "description": "Search query string"
      },
      "start_time": {
        "type": "string",
        "description": "The start of a time range to filter facts by their temporal qualifiers"
      },
      "end_time": {
        "type": "string",
        "description": "The end of a time range to filter facts by their temporal qualifiers"
      },
      "start_operator": {
        "type": "string",
        "description": "Operator for start time comparison (lt, le, ge, gt, eq)."
      },
      "end_operator": {
        "type": "string",
        "description": "Operator for end time comparison (lt, le, ge, gt, eq)."
      }
    },
    "required": ["query"],
  }
}
```

```
{
  "type": "function",
  "name": "find_gist_contexts",
  "description": "Explore related 'gists' (via synonym relationships) and connected 'facts' (triples) for a specific gist, with
optional temporal filters.",
  "parameters": {
    "type": "object",
    "properties": {
      "gist_id": {
        "type": "number",
        "description": "The index (not content) of a gist node from the last step to explore, starting from 1",
      },
      "start_time": {
        "type": "string",
        "description": "The start of a time range to filter facts by their temporal qualifiers"
      },
      "end_time": {
        "type": "string",
        "description": "The end of a time range to filter facts by their temporal qualifiers"
      },
      "start_operator": {
        "type": "string",
        "description": "Operator for start time comparison (lt, le, ge, gt, eq)."
      },
      "end_operator": {
        "type": "string",
        "description": "Operator for end time comparison (lt, le, ge, gt, eq)."
      }
    },
    "required": ["gist_id"]
  }
}
```

Figure 7: The tool descriptions for *lexical_retrieve* and *find_gist_contexts*.

```
{
  "type": "function",
  "name": "find_entity_contexts",
  "description": "This tool finds facts (triples) that match the given criteria. You must provide at least one of 'subject',
'object', or 'predicate'.",
  "parameters": {
    "type": "object",
    "properties": {
      "subject": {
        "type": "string",
        "description": "The subject of a fact, e.g., 'E1' in (E1, was born in, E2)"
      },
      "object": {
        "type": "string",
        "description": "The object of a fact, e.g., 'E2' in (E1, was born in, E2)"
      },
      "predicate": {
        "type": "string",
        "description": "Filter for specific relations by exact name (e.g., 'was born in')"
      },
      "start_time": {
        "type": "string",
        "description": "Filter facts based on their start time, e.g., '1952-01-01'. Use with start_operator to control
comparison"
      },
      "end_time": {
        "type": "string",
        "description": "Filter facts based on their end time, e.g., '1957-12-31'. Use with end_operator to control
comparison"
      },
      "start_operator": {
        "type": "string",
        "description": "Operator for start_time comparison. Options: 'ge' (>=), 'gt' (>), 'le' (<=), 'lt' (<), 'eq' (=). Default:
'ge'"
      },
      "end_operator": {
        "type": "string",
        "description": "Operator for end_time comparison. Options: 'ge' (>=), 'gt' (>), 'le' (<=), 'lt' (<), 'eq' (=). Default:
'le'"
      },
      "limit": {
        "type": "integer",
        "description": "Restrict the number of returned results. Use with 'ordering' to get the first/last items. Skip
setting this to get more contexts"
      },
      "ordering": {
        "type": "string",
        "description": "Order the results by time. 'asc' for ascending by start_time (earliest first), 'desc' for descending
by end_time (latest first)"
      },
      "offset": {
        "type": "integer",
        "description": "Skip the first N results. Use with 'ordering' and 'limit=1' to get the Nth item (e.g., 'the 2nd time')"
      },
      "aggregation": {
        "type": "string",
        "description": "Perform an aggregation and return a single number. Options: 'count', 'count_unique_subjects',
'count_unique_objects'."
      }
    },
    "required": []
  }
}
```

Figure 8: The tool descriptions for *find_entity_contexts*.

