# OpenReview forum: "REMem: Reasoning with Episodic Memory in Language Agent"
_ICLR.cc/2026/Conference — ICLR 2026 Poster_

### Official Review · Reviewer_7Ujh · 2025-10-27

**Soundness:** 3
**Presentation:** 2
**Contribution:** 2
**Rating:** 6
**Confidence:** 4

**Summary:**

The paper proposes REMem, a memory + reasoning framework for LLM agents. Offline, it converts past interactions into a “hybrid memory graph” containing (i) timestamped natural-language event summaries (“gists”) and (ii) structured temporal fact triples (subject, relation, object + validity interval). Online, a controller LLM acts as an agent: it issues tool calls to retrieve and traverse this graph (semantic retrieval, temporal filtering, entity-centric queries), iteratively gathers evidence, and then answers. The system targets both episodic recollection (“what happened when X visited?”) and episodic reasoning (“who lived there first?”, “how long did that last?”). REMem is evaluated on four benchmarks (LoCoMo, REALTALK, Complex-TR, Test of Time) and outperforms strong RAG-style baselines, structured memory systems (Mem0, Graphiti, HippoRAG 2), and even full-context prompting. On temporal reasoning, REMem-I is the only reported method to surpass 90% exact match on Test of Time, and it does so with lower average token cost than feeding the entire history to the model. The system also shows better-calibrated refusals on unanswerable questions.

**Strengths:**

1. Conceptual clarity of task focus. The paper explicitly distinguishes two abilities—episodic recollection vs. episodic reasoning—and builds the system to address both. This sharpens what “memory for agents” should mean.
2. Consolidation-style memory construction. While your paper targets episodic memory, I would argue that what your methodology does is more akin to systems consolidation in the neuroscientific sense. Systems consolidation in the brain entails that raw episodic memories are gradually distilled is distilled into schematic structures, which can be recalled in a compositional manner that may or may not include temporal details. This is a compelling stance that links agent memory to well-studied ideas in neuroscience. I think the paper would benefit from explicitly linking to this - I have left it as a strength as I believe it is one of the more interesting ways to interpret your work.
3. Agentic temporal reasoning. The agent can iteratively call tools that (a) pull semantically relevant gists and (b) query the graph with temporal/entity constraints (“who occupied Apt B before Casey moved in?”), enabling before/after, duration, and ordering queries that vanilla RAG typically fails at. This also ties in to compositional retrieval as seen in the brain.
4. Empirical results. REMem variants consistently outperform baselines on four datasets, including challenging temporal QA. The gains over “full context” are large, suggesting real reasoning benefits, not just better recall. The approach is also more token-efficient and shows more calibrated abstention (“I can’t answer”) than prior memory systems.
5. Ablations and human eval. The paper ablates removal of gists vs. fact triples and shows both are important, especially for multi-hop temporal reasoning. It also shows that its LLM-as-a-judge metric correlates better with human judgment than F1/BLEU, which strengthens trust in the reported numbers.

**Weaknesses:**

1. Connection to memory consolidation. As mentioned earlier, I would argue that your methodology specifically targets memory consolidation - reframing from this perspective would give a stronger theoretical connection to neuroscience.

2. Lack or discussion on parametric memory stores. In the introduction you mention that “parametric memory, embedded in modelweights during pre-training or fine-tuning, lacks adaptability and contextual grounding in specific experiences”. A whole breadth of literature covers parametric episodic memory stores that exist outside of the model weights. Some of these methods focus on restructuring internal model states (see [1]), others focus on event compression into a fast weight matrix [2]. I would argue that such methods are relevant and should be commented on/compared to as baselines.

3. Lack of mathematical/formal specification. The methodology reads descriptively but lacks a clear formalism. There is no precise definition of the memory graph schema (node/edge sets, attribute spaces, temporal operators), the retrieval operators as functions over that schema, or the agent loop as a constrained decision process. Even a lightweight formalization (typed multigraph with timestamped edge attributes and well-defined filter/composition operators) would greatly improve clarity and reproducibility.

4. Graph construction is under-specified. Even with the figure, it remains unclear:
    * whether gist nodes and phrase/concept nodes live in a single unified graph or in two stores queried separately;
    * where timestamps/intervals reside (on gist nodes, on relation edges, or on event-instance nodes) and how overlaps/validity are represented;
    * how entity canonicalization across mentions works (e.g., “I”→“Alice”; handling other arguments like “Peter”);
    * how synonymy links between gists interact with temporal distinctness (risk of over-merging similar but distinct events). Since retrieval operates over this graph, a precise schema is important.

5. Redundant first-stage retrieval functions. During initial recall the system exposes both a semantic retriever (embeddings) and a lexical retriever (BM25). Both serve the same purpose, yet the paper doesn’t justify why both are needed, how they’re combined, or quantify their individual contributions. If REMem benefits from simply having two parallel high-recall channels, that should be isolated and reported.

6. No end-to-end example trajectories. Given the centrality of a ReAct-style loop, the paper should print at least one full trace on a hard temporal query (sequence of tool calls, parameters, intermediate results, final answer). Without this, it’s hard to see how multi-step reasoning actually unfolds.

7. Breadth of evaluation. Most evaluation focuses on temporally framed questions (before/after, first/last, duration). It’s less clear how well REMem handles non-temporal personalization or causal/relational queries where “when” isn’t central, despite claims that gists capture richer situational context.

8. Memory complexity and system cost. The paper does not analyse space/time complexity of the memory store or retrieval: expected growth of gist nodes, phrase nodes, and relational edges with conversation length; frequency/impact of synonymy links; and the asymptotic/empirical cost of Stage-1 retrieval and graph exploration as histories reach hundreds of thousands or millions of tokens. A brief complexity discussion (e.g., storage per event, index sizes, average degree) plus empirical reporting (graph size, tool-call count, latency) would strengthen deployability claims.

[1] Z. Fountas, M. A. Benfeghoul, A. Oomerjee, F. Christopoulou, G. Lampouras, H. Bou-Ammar, and J. Wang, “Human-inspired Episodic Memory for Infinite Context LLMs,” arXiv:2407.09450 [cs.AI], 2025.

[2] M. Zhang, S. Arora, R. Chalamala, A. Wu, B. Spector, A. Singhal, K. Ramesh, and C. Ré, “LoLCATs: On Low-Rank Linearizing of Large Language Models,” arXiv:2410.10254 [cs.LG], 2025.

**Questions:**

* Can you provide a compact formal schema: node/edge types; where timestamps/intervals live; how contradiction and multi-interval states are represented?
* Why are both semantic (embedding) and lexical (BM25) retrievers needed in Stage-1 recall, and how are they combined? What happens if one is removed?
* Can you include one full ReAct trajectory on a representative complex query (tool calls, parameters, intermediate results, final answer)?
* Could you add a small slice evaluating non-temporal questions to demonstrate breadth beyond time-centric tasks?
* Please report storage and runtime characteristics—e.g., growth of gists/phrase nodes/edges with history length, index sizes, average node degree, effect of synonymy edges on sparsity/density, average tool-calls per query, and latency curves.

---

> ### Author Response · Authors · 2025-11-23
> **To Reviewer 7Ujh**
>
> We are grateful to the reviewer for emphasizing the conceptual clarity of our episodic focus and the strength of our empirical results, ablations, and human evaluation. For our work’s unique contribution, please refer to Response to All Reviewers, and we then address your detailed comments below.
>
> > Q1/W3: The formalism of REMem for graph structure, indexing, and inference.
>
> We’ve updated the detailed memory schema and algorithm procedure in Appendix B of our paper draft.
>
> > Q2/W5: Why are both semantic (embedding) and lexical (BM25) retrievers needed in Stage-1 recall? How are they combined?
>
> We provide both semantic and lexical retrieval tools because they serve different roles: embedding-based search is good for fuzzy, paraphrastic matches, while lexical is more precise and robust for exact entity names, dates, and rare tokens, which we found important for temporal and entity-centric queries.
> Rather than a fixed design, these tools are meant as pluggable components of the agent’s toolkit. At each step of tool invocation, the agent autonomously selects the appropriate retrieval strategy, e.g., using semantic retrieval in general cases while switching to lexical retrieval for specific entity identifiers.
> In the revision, we will add ablations to quantify their individual contributions.
>
> > Q3/W6: ReAct trajectory on a representative complex query
>
> The demonstration sections (highlighted in blue) of the prompts shown in Figures 3, 4, and 5 (Appendix D) present examples of each step.
>
> > Q4/W7: Could you add a small slice evaluating non-temporal questions to demonstrate breadth beyond time-centric tasks?
>
> Our evaluation on LoCoMo and REALTALK already includes both temporal and non-temporal questions. In Appendix C.7, we further introduce this evaluation perspective and report that the performance of REMem-I on LoCoMo non-temporal questions (75.0 LLM-J) is comparable to its performance on temporal ones (77.7 LLM-J), indicating that REMem is not limited to time-centric tasks.
>
> > Q5/W8: Please report storage and runtime characteristics
>
> Please refer to the updated Appendix of our paper draft: Appendix F.3 for system costs and Appendix F.5 for graph properties. We’ll add even more details for the next revision.
>
> > W1: Connection to memory consolidation.
>
> Thank you for pointing out the connection to memory consolidation. The indexing phase distills interaction into a more schematic hybrid structure, and the inference phase recombines these structures compositionally for reasoning, which parallels how systems consolidation abstracts and reorganizes episodic traces into more structured knowledge. We will revise the introduction and discussion to make this connection explicit.
>
> > W2: Lack of discussion on parametric memory stores.
>
> We thank the reviewer for pointing us to recent work on parametric episodic controllers such as EM-LLM and LoLCATs. In the introduction, our discussion of “parametric memory embedded in model weights” focuses on strictly weight-level storage. In the revision, we will broaden this discussion to also cover parametric episodic controllers that maintain state outside the static weights. As EM-LLM and LoLCATs require modifying and re-training the backbone model on long-context corpora, they are not straightforward baselines in our current non-parametric setting. Instead, we will position them as complementary parametric episodic approaches in Related Work.
>
> > W4: Graph construction is under-specified.
>
> Please refer to the details in Appendix B Formal Definition of our updated paper draft.

---

> > ### Comment · Reviewer_7Ujh · 2025-11-27
> > **Response to Authors**
> >
> > Thank you for your response and the amendments that you have already made thus far. I am satisfied with your responses, and would be happy to raise my score to an 8 once the full set of amendments have been made.

---

> > > ### Author Response · Authors · 2025-11-27
> > > **To Reviewer 7Ujh**
> > >
> > > Dear Reviewer 7Ujh,
> > >
> > > Thank you for your constructive feedback, which is crucial in encouraging us to revise the paper and release updates.
> > >
> > > As mentioned in the public comments *Update Notes for Paper Draft (2)*, the revisions we committed to in our initial response, particularly all **planned experimental parts, are now included in the updated PDF**.
> > >
> > > Sincerely, thank you for considering improving the score, and we will continue to refine the writing in other sections as well.

---

### Official Review · Reviewer_oAJd · 2025-10-30

**Soundness:** 3
**Presentation:** 3
**Contribution:** 3
**Rating:** 6
**Confidence:** 3

**Summary:**

This paper examines the memory capabilities of language agents, focusing on the challenges of episodic recollection and reasoning. To tackle these issues, the authors propose REMem, a two-phase framework for memory construction and utilization: (1) offline indexing, and (2) online inference. The offline phase converts experienced events into a hybrid memory structure consisting of (a) gists — concise representations that capture the key contextual details of each episode, and (b) facts — structured triples in the form of (subject, predicate, object). During online inference, the system retrieves and composes relevant information from the memory graph to support reasoning. REMem achieves significant improvements across various episodic recollection and reasoning benchmarks, demonstrating the effectiveness of explicitly modeling when and how events occurred. Additionally, the method enhances refusal behavior by reducing hallucinations when the necessary information is missing.

**Strengths:**

- **S1.** REMem provides a well-motivated and well-structured approach to episodic memory, explicitly incorporating temporal and contextual information that prior non-parametric memory systems typically lack.
- **S2.** The hybrid structure of gists and facts, the method for constructing this structure, and the use of multiple retrieval tools to enable multi-step reasoning together represent a novel contribution.
- **S3.** The paper evaluates the model on both episodic recollection and episodic reasoning tasks, demonstrating significant performance improvements. Additionally, the authors conduct a comprehensive analysis, including refusal behavior, error analysis, and human evaluation, which provides deeper insight into the strengths and limitations of the method.

**Weaknesses:**

- **W1.** Since gist and fact extraction rely heavily on LLMs, errors introduced during this process may propagate and negatively impact the entire REMem framework. In particular, gist extraction is an abstractive procedure that can be prone to hallucination, making error detection increasingly important as the memory store expands.
- **W2.** The distinction from prior KG-based memory systems (e.g., HippoRAG’s offline indexing and online retrieval pipeline) is not clearly articulated. While the incorporation of temporal modeling is highlighted as a contribution, further elaboration would help readers better understand the core novelty and value of the proposed approach.
- **W3.** The paper would benefit from more detailed descriptions and concrete examples of how its core components — such as retrieval tools and graph traversal — operate together during inference. Additional clarification could significantly enhance the reader’s understanding of the system’s practical workflow (see Q1–Q3).
- **W4.** (Minor formatting issue) In lines 211–215, the numbering format repeats “1)” twice. Using alternative markers such as “(a)” and “(b)” would improve readability.

**Questions:**

- **Q1.** How are synonymy edges between gist nodes utilized in retrieval or reasoning? Do they improve multi-hop recall or only expand local connectivity?
- **Q2.** In the retrieval tools, are semantic_retrieve and lexical_retrieve applied to both gist and fact entities? If so, does this imply that gists function at the sentence level, while facts operate at the keyword/entity level?
- **Q3.** During graph exploration, once a relevant node is identified, are both its associated gists and facts incorporated into the reasoning workflow?
- **Q4.** Fact extraction employs a schemaless approach for predicate construction. How does the system handle inconsistency in predicate expressions (e.g., synonymy, phrasing differences)?
- **Q5.** When gists or facts extracted by the LLM contain errors, how does REMem detect these issues, and what strategies could be used for correction or quality improvement?

---

> ### Author Response · Authors · 2025-11-23
> **To Reviewer oAJd**
>
> We thank the reviewer for acknowledging the motivation, structure, and empirical effectiveness of REMem, as well as the value of our analysis on refusals and errors. For an overview of the core contributions and the relation to prior KG-based memory work, please see our Response to All Reviewers, followed by our answers to your specific questions.
>
> > Q1: How are synonymy edges between gist nodes utilized in retrieval or reasoning?
>
> In REMem, synonymy edges between gist nodes are used during graph exploration as additional hops that let the agent move from one episode to nearby, semantically similar episodes, which effectively expands local connectivity and improves multi-hop recall.
> We will clarify this role in the main text and add an ablation comparing REMem with and without synonymy edges to quantify their impact on multi-hop reasoning.
>
> > Q2: In the retrieval tools, are semantic\_retrieve and lexical\_retrieve applied to both gist and fact entities? If so, does this imply that gists function at the sentence level, while facts operate at the keyword/entity level?
>
> Both retrieval tools can query gists and facts, and both gists and facts are used in the reasoning workflow.
> Gists function at the sentence level, and facts operate at the phrase level.
> Both of your assertions regarding tools and granularity are accurate interpretations, though they represent distinct dimensions without a causal relationship.
>
> > Q3: During graph exploration, once a relevant node is identified, are both its associated gists and facts incorporated into the reasoning workflow?
>
> Once a relevant context is retrieved, either gists or phrases (in triplet form), the agent incorporates both its associated gists and phrases into the reasoning process.
> Retrieval, graph exploration, or QA reasoning in each iteration is designed as a hybrid view for the hybrid graph without the need to worry about deciding which type of context to explore.
>
> > Q4: How does the system handle inconsistency in predicate expressions during extraction?
>
> We currently adopt a schemaless design for predicates and do not perform explicit canonicalization across different surface forms. In our experiments, retrieval is mainly driven by phrases, temporal constraints, and textual semantic similarity over gists and facts, so we did not observe a clear, systematic degradation that we could attribute solely to predicate variation. That said, we agree that learning or inducing a shared predicate schema for larger and more heterogeneous corpora is an important and underexplored direction, and we will clarify this limitation and point to it as future work.
>
> > W1/Q5: How does REMem detect extraction errors?
>
> We do not perform explicit error correction during the extraction process.
> However, we addressed a closely related question (f9y5 Q2) concerning the robustness of extraction: Based on LLM quantitative analyses and human qualitative evaluation, we believe that our extraction outcomes surpass prior work and leave error correction as future work.
>
> > W2: The distinction from prior KG-based memory systems? E.g., HippoRAG.
>
> Unlike prior KG-based memory systems such as HippoRAG, REMem does not rely on a fixed PPR-style random walk over a graph, but exposes the memory graph as tools that the agent composes into multi-step processes.
> PPR can rank nodes that co-occur with a query, but it fundamentally cannot express directional, interval-based constraints like “who lived there before Alice moved in?” or “how long did Alice stay?”, so episodic reasoning is left implicit to the LLM on a flat neighborhood.
> In REMem, episodic operators are first-class in the graph (gists and fact intervals) and in the agent’s tool calls, and our experiments on both episodic recollection and reasoning show consistent gains over HippoRAG2, indicating that this explicit episodic reasoning layer is empirically useful beyond generic KG-based retrieval.
>
> > W3: The detailed descriptions and examples for the inference phase?
>
> The demonstration sections (highlighted in blue) of the prompts shown in Figures 3, 4, and 5 present examples of each step. For Q1-Q3, please see above.
>
> > W4: Minor formatting issues (line 211-215).
>
> Thank you for pointing it out. We’ve formalized it better.

---

### Official Review · Reviewer_f9y5 · 2025-10-30

**Soundness:** 2
**Presentation:** 4
**Contribution:** 2
**Rating:** 4
**Confidence:** 4

**Summary:**

This paper proposes REMem, a two-phase episodic memory system for language agents: (1) offline indexing that converts experiences into a hybrid memory graph containing time-aware gists and structured facts, and (2) online agentic inference using carefully curated retrieval and graph exploration tools. The authors evaluate on four benchmarks spanning conversational QA (LoCoMo, RealTalk) and temporal reasoning (Complex-TR, Test of Time), showing 3.4% and 13.4% absolute improvements on episodic recollection and reasoning tasks respectively over three SoTA systems (Mem0, Graphiti, and HippoRAG 2).

**Strengths:**

- Presentation: Very well motivated and very well written paper (clear problem framing, separating episodic recollection from episodic reasoning and evaluates both). Figures are also nice and helpful.

- Good execution: solid engineering, reasonable baselines, ablations that show both gists and facts matter, and an efficiency comparison. Refusal analysis is an added bonus.

**Weaknesses:**

- (main concern) The paper evaluates on relatively controlled settings where proper knowledge extraction is assumed to work. The most critical and challenging aspect of the work (how robustly the gist/fact extraction generalizes to noisy, ambiguous, real-world text) receives minimal treatment in the paper. Once you have clean, well-structured graphs, the superior performance on episodic reasoning tasks becomes somewhat predictable rather than surprising. Unfortunately, the paper spent most of its efforts narrating a success story on this predictable part. As said earlier, the ablation studies are helpful but don't address extraction robustness.

- The graph+LLM combination is now ubiquitous in LLM research. Numerous commercial and academic systems combine knowledge graphs with RAG/memory, including commercial products, and including for temporal reasoning. What makes REMem's specific formulation necessary or superior isn't sufficiently differentiated from this crowded landscape, leading to a feeling of : yet another LLM + graph paper.

- The paper rightfully motivates the need for episodic reasoning, and uses four datasets. However, although they do have episodic characteristics, none of these datasets was originally designed for episodic memory. Other recent benchmarks could strengthen claims about the method's episodic capabilities and generalization. e.g. (keyword "episodic memory benchmark" yields:
https://arxiv.org/abs/2501.13121
https://huggingface.co/papers/2410.08133 )

- The code is not available yet (written: upon acceptance)

**Questions:**

- A plethora of approaches mix knowledge graphs and RAG/memory retrieval in LLMs, including commercial products (graphrag, entigraph by memory but keyword "Graph LLM" "Graph LLM memory|rag" yieds much more), including temporal reasoning. Only a tiny subset is mentioned in the related work. Why were these methods omitted?

- How robust is extraction to real-world noise? What percentage of extractions are incorrect or incomplete? How does performance degrade with noisier inputs? Can you provide extraction accuracy metrics on held-out data? How does the extraction perform on regular text outside the 4 datasets you tested?

- perhaps for the discussion section: LLMs succeeded where decades of knowledge graph methods failed, largely because their learned representations are more flexible than rigid symbolic structures. Why should we return to explicit graph extraction? The paper doesn't engage with this question. Is the implicit claim that episodic memory specifically benefits from explicit temporal structure?

---

> ### Author Response · Authors · 2025-11-23
> **To Reviewer f9y5 (1)**
>
> We appreciate the reviewer’s positive assessment of the problem framing, the clarity of the writing, and the solid engineering and ablation studies. For a general view of how REMem differs from prior graph-based memory systems, please see our Response to All Reviewers before our following specific replies.
>
> > Q1: Why were other commercial/academic systems omitted in Related Work?
>
> We appreciate your concern regarding the fairness and comprehensiveness of our experimental setting and related work discussion.
> As we indicated in the Introduction, we analyzed the issues present in common paradigms and focused our experiments on recent, open, and strong representatives such as Mem0, Graphiti, and HippoRAG 2\. In particular, the HippoRAG 2 paper reports clear gains over prior GraphRAG-style systems, so outperforming HippoRAG 2 already places REMem against a very competitive baseline.
> Regarding references to other commercial or academic approaches, we apologize that we were unable to provide more comprehensive coverage due to space constraints, and we will incorporate a broader discussion and citations to these systems in the revised version.
>
> > Q2: How robust is extraction to real-world noise?
>
> We hope that the following LLM-based quantitative analysis and manual qualitative analysis are sufficient to address your concerns regarding extraction quality.
>
> **LLM-based Quantitative Analysis**
> We randomly sampled 50 generated gists and 50 facts each from LoCoMo, REALTALK, and MuSiQue, and used an LLM to judge whether each gist or fact satisfies faithfulness and is free of hallucination. The results are categorized into five levels:
>
> * **5:** Completely accurate and appropriately specific.
> * **4:** Mostly accurate, with only very minor issues.
> * **3:** Mixed: some correct content, but also clear issues.
> * **2:** Largely inaccurate or misleading, with only a few correct details.
> * **1:** Completely wrong, unrelated, or nonsensical.
>
> Please note that these three datasets are included in our experiments (also Appendix C.5). LoCoMo is synthesized by LLMs, REALTALK is collected from **real** **human** **conversations**, and MuSiQue is built from **widely** **used** **Wikipedia** content. Therefore, the data we rely on is diverse and not detached from so-called \`real-world data\`. The summarized results are as follows. As can be seen, the vast majority of extracted results are judged to be faithful and free of hallucinations, with no significant errors in extraction.
>
> | Scores (Ratio) | LoCoMo | REALTALK | MuSiQue |
> | :------------- | :----- | :------- | :------ |
> | 5              | 92%    | 91%      | 97%     |
> | 4              | 6%     | 8%       | 2%      |
> | 3              | 2%     | 0%       | 1%      |
> | 2              | 0%     | 0%       | 0%      |
> | 1              | 0%     | 1%       | 0%      |
>
> Information extraction alone is a complex research topic. We avoid using complicated schemas and instead leverage LLMs from both the concept level and the context level to obtain relatively high-quality extraction results.
>
> **Qualitative Analysis**
>
> In updated Appendix F.4, we present examples for extraction results. It is evident that compared with Mem0, REMem yields more comprehensive and organized results, and it can even infer implicit temporal information, such as the end of a game season.
>
> > Q3: LLMs succeeded where decades of knowledge graph methods failed. Why should we return to explicit graph extraction?
>
> Thank you for raising this insightful point about the role of explicit graphs in the era of powerful LLMs. We agree that modern LLMs excel at capturing broad, general-domain knowledge in their parametric weights, and that graph-based memory and RAG-style methods are now ubiquitous. Our position is that implicit, parametric representations in LLMs and explicit, non-parametric memory structures are complementary rather than competing.
> In REMem, explicit, time-aware structure serves to mitigate key limitations of purely latent representations for episodic memory: it enforces ordering and interval constraints through deterministic operators, enables targeted multi-hop composition over long and heterogeneous histories, and yields interpretable evidence that supports conflict detection and more calibrated abstention. Meanwhile, the LLM retains the role of abstraction and paraphrasing over this structure. More broadly, we view REMem as an external memory module that goes beyond a single set of pretrained weights: it personalizes the agent’s behavior to a particular stream of experiences, supports continual updates without retraining, and helps mitigate forgetting.
> We will clarify this complementary view of parametric and non-parametric memory, and better position REMem within the broader landscape of graph-based memory and RAG systems in the revised version.

---

> ### Author Response · Authors · 2025-11-23
> **To Reviewer f9y5 (2)**
>
> > W1: The paper evaluates in relatively controlled settings where proper knowledge extraction is assumed to work. Once you have clean, well-structured graphs, the superior performance on episodic reasoning tasks becomes somewhat predictable rather than surprising.
>
> First, as we have clarified in the above response to Q2, REMem can extract information reliably from realistic conversational or semantic texts, which is not based on any assumption but on both LLM-based and human **evaluations**. In particular, evaluation on REALTALK, which consists of human dialogues containing abundant colloquial expressions and noise, shows that our improvements over prior work are even greater than on the synthetic LoCoMo dataset (Table 3). This indicates that our evaluation is not restricted to overly controlled or “clean” settings.
>
> Second, even though the specific Test of Time dataset is relatively clean and structured, strong performance on episodic reasoning is by no means predictable. The challenges in constructing non-parametric long-term memory for language agents are **multifaceted**, including representation and extraction, retrieval, and reasoning. Existing RAG or non-parametric approaches do not perform well even under such simpler encoding or extraction conditions. In contrast, REMem achieves a substantially larger advantage and is the only method that surpasses 90% EM owing to its appropriate memory representation and strong agentic inference module, which suggests that superior performance is not “predictable” once clean graphs are available.
>
> > W2: What makes REMem's specific formulation necessary compared to other Graph \+ LLM systems?
>
> Please refer to REMem’s Contribution in Response to All Reviewers.
>
> > W3: What is the rationale for selecting these benchmarks?
>
> Thank you for your valuable suggestions regarding benchmark selection. We’ll complete the references and discussion regarding benchmarks.
>
> We believe academic benchmarks for episodic memory remain in an early stage of development, regardless of whether papers explicitly claim “episodic memory” in their titles, and the number of available benchmarks is scarce. Existing benchmarks typically cover only specific data formats, such as conversations, long stories, or encyclopedic entries, and may be limited in difficulty.
>
> Therefore, we selected four benchmarks across both episodic recollection and episodic reasoning for a relatively comprehensive evaluation. Such integrated evaluation is uncommon in prior work, where many approaches focus only on a subset of a conversational benchmark or on specific semantic QA tasks.
>
> In addition, the LoCoMo and REALTALK benchmarks we chose naturally cover sequential comprehension questions similar to SORT (https://huggingface.co/papers/2410.08133), and even more additional challenges. Please see the updated Appendix C.7 for our discussion of multiple temporal challenges.
>
> > W4: The code is not available yet.
>
> As publicly committed, the code and data will be open-sourced upon acceptance.

---

> > ### Comment · Reviewer_f9y5 · 2025-11-27
> >
> > I thank the authors for the clear rebuttal and the friendly responses, and also for the efforts of adding the LLM judgments. I still think the paper is well executed and beautifully presented. The empirical improvements over baselines are reproducible and meaningful within the chosen evaluation setup.
> >
> > Sorry, however, for failing to be more explicit with what I meant with real world data and real world episodic events. Humans rarely get the explicit timestamps of events, and when they get them, they are scarce. This is also reflected in the narratives they create and which LLMs would have to eventually process.
> >
> > Please correct me if I'm wrong but the (episodic not semantic) benchmarks that you considered (e.g. ToT, Realtalk, LoCoMo) are artificially "time-rich" in ways that simplify the indexing challenge (and inference of absolute time from relative cues). I acknowledge that REALTALK might reflect (some) realistic scenarios where system timestamps are available as metadata, but such anchors would be sparser or more distant from their events. Hence, current benchmarks contain REMEM-friendly information that is "ready to be easily indexed" (hence no surprise that the indexing will work and the performance will improve). My question remains: what happens when dates are implicit or when they are very far apart in the text from the events of interest. How would one build a generic episodic index that generalizes beyond timestamp-adjacent retrieval? or even how to augment long conversational with the temporal dimension.
> >
> > Example from ToT (where it does not seem surprising that a system designed to parse temporal expressions and store them as indexed facts should get 90% on this):
> > ```
> > Here is a set of temporal facts: E57 was the R14 of E83 from 1959 to 1963. E11 was the R35 of E26 from 1959 to 1961. E43 was the R35 of E94 from 1959 to 1963.
> > ```
> >
> > Now, I recognize that ToT was designed for controlled evaluation of temporal reasoning, and REMem. However, this plus the fact that you're still transforming unstructured text into structured representations (an approach shared by a plethora of other works) gave me the concern that the contribution may be more incremental than transformative. While the specific combination of gists, time-scoped facts, and agentic retrieval is carefully designed, it reads as a well-executed engineering contribution, with a solution that was designed to naturally work well for the benchmarks where temporal information is abundantly and closely accessible.
> >
> > I remain uncertain about acceptance and really hope that the authors will prove me wrong in the future, showing how REMem or its successors can actually solve real-world episodic memory challenges: reconstructing timelines from long novels  where temporal anchors appear pages away from the events they govern, inferring event order from causal cues rather than explicit dates, and finally the ultimate goal: making LLM conversations last for hours without context rot, and enabling longer agent interactions.
> >
> > Sorry if I misunderstood your work, I hope my point is clearer now. I'm still open to hear your arguments and equally open to change my mind.

---

> ### Author Response · Authors · 2025-11-30
> **To Reviewer f9y5 (3)**
>
> Dear Reviewer f9y5,
>
> Thank you again for the thoughtful clarification. We genuinely appreciate that you are pushing toward a much broader notion of “real-world episodic memory,” and we agree that the long-term vision you describe, like novels with sparse temporal anchors, is both important and still unsolved for the community.
>
> ### **What counts as truly difficult episodic memory?**
>
> We would like to note that explicit temporal annotations do not undermine the substantive difficulty of the episodic reasoning problem we target.
>
> Episodic memory is challenging in many dimensions, and the temporal dimension in particular remains complex. It requires the model to understand the intricacies of context, identify and interpret temporal cues, and perform multi-step reasoning over them.
>
> Although we don't believe existing benchmarks are perfect testbeds of episodic memory, **we attempted to formulate the tasks** as episodic recollection and reasoning, and to adopt a representative set of benchmarks that together provide **a relatively comprehensive evaluation suite**.
>
> Comprehensive experiments show that these challenges are not trivial for existing systems: strong RAG and structured-memory baselines remain far below REMem on episodic reasoning and calibrated refusal, despite receiving the *same* time-rich input. If the tasks were easy merely because timestamps are adjacent, then these baselines should already saturate performance, which is not what we observe.
>
> Our contribution is not to show that “date parsing works,” but that a hybrid episodic-graph representation combined with agentic retrieval yields substantial and reproducible improvements over those baselines under a well-defined episodic-memory setting.
>
> ### **On “real-world data” vs. benchmark data**
>
> We would also hesitate to say that any existing benchmark, including the ones you kindly suggested, uniquely captures real-world data. All available benchmarks, including the ones we used, are necessarily designed abstractions that highlight certain aspects of reality.
>
> The scenarios we study are closer to logs and conversational traces where system-level timestamps or temporal expressions are part of the environment (e.g., chatbots, personal assistants). The benchmarks you highlight lean more toward narrative text with sparser anchors. Both families are valuable, but they target *different* slices of the real-world space.
>
> ### **Episodic memory for language agents**
>
> Our focus is on building stronger memory, particularly episodic memory, for language agents. For agentic tasks, it is natural to assume access to temporal metadata: operating systems, browsers, messaging platforms, and logging infrastructure all maintain precise timestamps and session boundaries. From an agent-centric perspective, many practical episodic-memory scenarios *do* look “time-rich”: the challenge is not whether the agent can see time, but how to turn a long stream of time-stamped experiences into a usable episodic structure that supports compositional reasoning and safe refusal.
>
> For the case where most temporal anchors are invisible and the mode is required to infer time solely from subtle causal or discourse cues, the problem shifts toward temporal reasoning and long-context understanding, which is an important but *distinct* line of work. We fully agree that such settings are valuable, while our claim in this paper is more modest and more targeted: given that language agents in many applications *do* have access to temporal background information, REMem is an effective way to encode and use that information for episodic recollection and reasoning.
>
> ### **A simple sanity check of REMem on EPBench**
>
> Given the above reasons, we believe the current work is sufficiently complete as a full paper, and we encourage future research on episodic memory, including both the new benchmarks and the new methods. Nevertheless, we conducted a simple evaluation of **REMem-I** (GPT-4.1-mini + NV-Embed-v2) on the benchmark you mentioned, **EPBench** (https://github.com/ahstat/episodic-memory-benchmark), as a sanity check.
>
> According to an LLM judge, our system answers around **50%** of the questions correctly on the EPBench generated with GPT-4o (the context is ~13,680 tokens). REMem without any task-specific design achieves the equal performance as the NV-Embed-v2 RAG baseline. Considering that many LLMs reported in the benchmark achieve **below 30% accuracy**, we believe REMem delivers a reasonable level of performance.
>
> Please note that, based on our observations, EPBench is largely constructed from LLM-generated data. We view this evaluation as a sanity check rather than evidence of real-world generalization.
>
> Thank you again for posting this important question.

---

### Official Review · Reviewer_1gLe · 2025-10-31

**Soundness:** 3
**Presentation:** 3
**Contribution:** 2
**Rating:** 6
**Confidence:** 4

**Summary:**

The authors introduce a framework named REMem for indexing and reasoning with episodic memory. A two step approach is proposed: first, an *offline indexing* convert experiences to a time-aware memory graph, then an agentic retriever (in a single step or in iterative steps) performs the *online inference*.

In the offline phase, a memory graph is created (from the "event statements") containing both: *gists*, that are event summaries with timestamp, and *facts*, that (subject, relation, object) triplets with additional time information (date or range). Example of gist "Alice bought 3 cows from Peter on Jan 17th" with the associated timestamp (17 Jan, 2024). Example of fact:  (subject Alice, relation purchased, object 3 cows)

In the online phase, the graph enables logic composition (e.g., time range filtering, neighbor exploration). An agent retriever uses retrieval, graph exploration and flow control tools to extract the answer of the question.

The framework is evaluated on 4 benchmarks including LoCoMo and Test of time, and compared against Mem0 and HippoRAG2 among others, on 2 tasks (recollection and reasoning). The experiments show that the methodology outperforms Mem0 and HippoRAG2 for both tasks, especially for refusal behavior and is on-par compared to full context performance.

**Strengths:**

- Grounding the events in time by design (compared to e.g., Mem0)
- Better performance over HippoRAG2 and Mem0 for most experiments
- Error analysis explanation in Sec. 6.4 regarding the remaining gap of performance in this experiment
- Correct evaluation metrics

**Weaknesses:**

- Limitations have not been highlighted
- No conflict detection and delete/update, compared to Mem0. This is not discussed at all in the paper
- Experimental settings can be discussed, in particular the chunking / short size of each session
- Modeling of events with grounding in time, but other aspects like spatial location are not systematically modeled
- No data/code provided at submission time

**Questions:**

My current grading for the contributions is "fair". I may reassess this aspect after discussing the limitations of the experimental settings, that are currently not discussed by the authors. In particular, the limitations of the work are only discussed w.r.t. the current experiment (like error analysis). But the main limitations regarding "more complex environments" (l. 486) are never explained:

- the chunking is simplistic (well segmented short chat sessions): the chunking of each event (event statement, chat session) is given by the benchmark, but in realistic scenario it might not be the case: the absolute date may not be located in the current chunk or linked to the current date (it may be implicit or not available); same for location, or other entities. Relative temporal expressions may not be resolved locally (while in the paper, the model seems to be forced to give an absolute date).

Q. what are the assumptions needed on the input data ("event statement") for ensuring the correctness of the methodology? How the "event statement" have been obtained/defined?

Q. the entities e.g. "Alice" can be recurring. How this is handled? Is there a risk of confusion for large graphs? Can the model grounds two different "Alice"? How to resolve conflicts? Another example can be for referring to a book and a movie sharing the same name but with slight adaptation.

- in-context setting gives the same performance.

Q. If I understood correctly, all the documents are integrated in a single large graph (that might be composed of multiple non connected subgraphs). Can you confirm (and highlight) that for building the graph, you consider one for the whole corpus of experiences (instead of one separated graph per chat session)?

Q. Can you describe the graph obtained? How many entities, how many links? Is it mainly a single connected graph (recurring entities), or an union of many disconnected graphs? Is it related to Table 2 (present but not referenced in the paper)?

Q. l.262: I don't get: "Due to its limited context window", what is it? If you are able to perform the experiment, why the context window is a limiting factor?

Q. Can you show a table with the time taken for building the offline graph as a function of the dataset size, and the time/cost for inference?


## Clarification questions and minor comments:

Q. Can you show a more complex facts graph in the appendix? The Fig. 2 represents two (s, r, o) relations. Is it possible for an object (like "3 cows") to be the subject in a following session? Is it possible to retrieve information from the object instead of the subject?

Q. l.200 what is "phrase node" there?

Q. What is the impact of the synonymy edges? (there is no ablation with and without it)? Is there a risk of confusion between events when the synonymy edges parameter increases?

I agree that on this experiment, the results are better overall, from the table, compared to Mem0 and HippoRAG2 for recollection.

Q. Does TISER or -I strategies applicable for HippoRAG2, for which performance seems comparable in Tab. 4 (against REMem-S)

---

> ### Author Response · Authors · 2025-11-23
> **To Reviewer 1gLe (1)**
>
> We are grateful to the reviewer for highlighting our temporal grounding, correct evaluation metrics, and the strength of our error analysis. For a consolidated discussion of REMem’s contributions, please refer to our Response to All Reviewers, after which we respond to your detailed questions below.
>
> > Q1: Limitations of experimental settings.
>
> Thank you for your thoughtful question. For “more complex environments” (line 486), we refer to highly challenging settings such as Web browsing or coding, in which research on agent memory remains underexplored.
>
> Though we did not include a standalone section titled “Limitations”, we discussed our method’s weaknesses in several parts of Section 6 (Discussions). Specifically, better but imperfect refusal behavior (§6.2), grounding or temporal/numerical reasoning mistakes (§6.4), and misunderstanding in long contexts compared to the RAG baseline (§6.5).
>
> > Q2: Limitations of chunking events.
>
> In our experiments, we simply reuse the session boundaries provided by the benchmarks for fairness and comparability. Thank you for pointing this out, and we view the design of chunking strategies as scenario-dependent and an important topic for future work. Based on our experiences, we believe that splitting long interaction streams into shorter and manageable sessions should be a natural and intuitive engineering choice, and our results indicate that REMem can already operate reliably under this setting.
>
> > Q3: Discussion about the availability of time expressions.
>
> In an agentic scenario, each interaction naturally comes with a timestamp and a session boundary from the environment rather than from the text itself.
> Meanwhile, REMem is not restricted to such cases where explicit absolute dates are present. In our paper, we evaluate **episodic** **reasoning** (Table 4\) settings where the challenge is about relative temporal relations or references to time.
> In these tasks, the model must infer temporal relations purely from relative cues rather than any absolute dates. Despite the increased difficulty, REMem still shows clear improvements over strong baselines, indicating that our memory graph and retrieval procedure do not rely on clean, well-segmented sessions with explicit timestamps and can operate effectively when time is only implicit or relative.
>
> > Q4: How to define \`event statement\`?
>
> In our setting, an event statement is simply a self-contained natural-language fragment that describes one or more concrete episodes. In practice, this corresponds to a short chat session or any textual segment where an event is expressed (e.g., “Alice bought three cows from Peter on Jan 17” in Figure 1).
>
> > Q5: How does REMem resolve conflicts like the recurring entity name?
>
> REMem does not perform explicit conflict *deletion* or hard resolution at indexing time. Instead, we preserve all extractions with timestamps and provenance and let the online agent reason over time.
> The rationale is that enforcing strict conflict resolution across all storage is extremely hard, and so-called conflicts are mostly issues of semantic memory with contradictory facts. For the episodic memory we focus on, we deliberately choose to retain distinct experiences. This design is intuitive and practical: similar to version control in coding, Git preserves the full history of diverging commits instead of overwriting them. Any useful code that appeared at some point in history can later be revisited, reused, or combined, and we view episodic traces in REMem in the same way.
>
> > Q6: Why is the context window a limiting factor for the Full-Context method (line 262)?
>
> Thank you for pointing out the confusion. Our point on line 262 is that, as an intuitive approach, Full-Context always has a limited context window. Therefore, it cannot serve as a memory approach, and we only use its performance as a reference. We have updated the relevant statements in our paper.
> In addition, the full-context approach suffers from high computational cost, and it is far inferior to REMem on episodic reasoning tasks, achieving 79.7 vs. our 93.1 EM on the Test of Time.
>
> > Q7: Does the scope of each graph cover a single session or multiple sessions?
>
> Yes, our implementation can build one unified graph for the entire corpus, rather than a separate graph per session. The exact granularity follows the design of each testbed:
>
> \- Conversational QA (LoCoMo / RealTalk): Each user profile corresponds to one memory graph. A user may have multiple sessions, and all their sessions are integrated into the same graph.
> \- Complex-TR: All event statements in the corpus are combined into a single global graph, since the benchmark is defined over a shared narrative.
> \- Test of Time: Each query comes with an isolated context, so we construct one graph per query, as the data do not share cross-query episodes.
>
> These choices match the structure of different scenarios, and the corresponding statistics are summarized in Table 2\.

---

> ### Author Response · Authors · 2025-11-23
> **To Reviewer 1gLe (2)**
>
> > Q8: Can you describe the graph obtained?
>
> Please refer to Appendix F.5, where we have updated some numbers about graph properties.
>
> > Q9: What are the costs of building graphs and inference?
>
> Please refer to updated Appendix F.3 Token Usage to see system costs. We’ll update the time costs in our revised version.
>
> > Q10: More examples of the graph.
>
> The demonstration sections (highlighted in blue) of the prompts shown in Figures 3, 4, and 5 present examples for each step.
>
> > Q11: What is a phrase node (line 200)?
>
> A phrase node represents a concept-level span from a factual triple (subject, predicate, object), either subject or object. Each phrase node denotes a participant, action, or object in an event and may carry attached temporal qualifiers.
> Please refer to the updated Appendix B Formal Definition for more details about the graph.
>
> > Q12: What is the impact of synonymy edges?
>
> We’ll show some ablation performance about this in our revised version.
>
> > Q13: The comparison with HippoRAG 2? Could HippoRAG 2 use TISER or \-I?
>
> 1\. Comparing HippoRAG is not an intuitive or trivial task. HippoRAG is a RAG method that retrieves from source information and then generates answers, while REMem (also Mem0 and Graphiti) retrieves from rewritten information. It may be more difficult to exploit the original source text than rewritten, higher-density content. To avoid downplaying HippoRAG 2’s performance, we feed three full sessions as context for each query. In practice, this gives HippoRAG 2 **an order of magnitude more tokens** than REMem-S (top-10 gists and facts), which is precisely what it needs to achieve reasonable performance in our setting.
>
> 2\. Even when provided sufficient context, HippoRAG 2 fails to outperform our method across all datasets, whether REMem-S or REMem-I. The gap is 9.5 LLM-J score on REALTALK and 26.2 on Test of Time.
>
> 3\. Regarding iterative retrieval, HippoRAG 2's PPR algorithm essentially performs multiple graph iterations within a single retrieval call, similar to a random walk process. Therefore, we consider it reasonable to compare it against both REMem-S and REMem-I directly.
>
> > W1: What is the modeling of other aspects, like spatial location?
>
> While time is essential for episodic memory, REMem is designed as a situational memory model. In the indexing phase, gists are prompted to capture a richer situation description, i.e., participants, spatial location, and other important contextual attributes, rather than only the timestamp. These attributes are expressed within the gists and facts and can be accessed through exactly the same retrieval tools.
>
> For general non-temporal questions, our evaluation on LoCoMo and REALTALK already contains both temporal and non-temporal questions.
> We add further evaluation on temporal and non-temporal questions in Appendix C.7 and show the performance of REMem-I on LoCoMo non-temporal questions (75.0 LLM-J) is on par with the temporal ones (77.7 LLM-J).
>
> > W2: Will the data and code be open-sourced?
>
> As publicly committed, the code and data will be open-sourced upon acceptance.

---

> ### Author Response · Authors · 2025-11-27
> **To Reviewer 1gLe (3)**
>
> Dear Reviewer 1gLe,
>
> Thank you for your constructive feedback, which is crucial in helping us to revise the paper.
>
> As mentioned in the public comments Update Notes for Paper Draft (2), the revisions we committed to in our initial response, particularly **all planned experimental parts, are now included in the updated PDF**.
>
> We hope these updates will help you evaluate the paper. We will also continue to refine other sections of the writing. Thank you!

---

> > ### Comment · Reviewer_1gLe · 2025-11-28
> >
> > Thank you for the clarifications and for the additional updated content. Based on the revised version, I will update the contribution score from 2 to 3 (for some reason the review cannot be edited currently).

---

### Official Review · Reviewer_JsEd · 2025-10-31

**Soundness:** 2
**Presentation:** 3
**Contribution:** 2
**Rating:** 4
**Confidence:** 4

**Summary:**

the paper presents REMem, a prompt-based approach for instructing LLMs encoding episodic memory as a graph of ``gists'' and facts; indexing is performed offline and is queried at inference with either a single-step or iterative procedure. (This decoupling raise serious concenrs on the usefulness of the approach)

author compare on 4 datasets against a set of alternatives (essentially,
Mem0, Graphiti and HippoRAG) and naive full-context/oracle baselines.

analysis is quite detailed, with analysis of errors and comparison of 100 samples to LLM as a judge answers, though it lacks statistically relevant depth.

summarizing, the paper tackles an important problem. it offers a relatively simple  solution (some doubts remain concerning practical relevance), and performs some experimental evaluation, with good  (deep dive, error analysis, human comparison)  as well as bad (lack f statitical rigour) aspects -- overall, the paper is therefore in a borderline position

**Strengths:**

- balanced comparison with 4 datasets and 3 non-naive baselines
- sufficiently detailed analysis of results

**Weaknesses:**

(check details in the question section)

- an apparent oximoron?
- lightweigth contribution (with apparent oximoron?)
- lack of statistical rigour in the analysis
- doubtful numbers reported, with lower than publicly reported baselines

**Questions:**

the paper tackles an important problem. it offers a relatively simple  solution, and performs some experimental evaluation.


## an apparent oximoron?

two main comments

- on the one hand,  the paper claims the contribution to be to "directly instruct the LLM" to create memories. I am not sure this is a real benefit, as at the end of the day, this is created through a prompt -- whereas an algorithm could be analyzed under some hypotesis on the distribution/properties graph of gists/facts, a prompt-algorithm through an LLM can only be observed through its output.

- on the other hand, the paper claims this indexing is still an offline phase, which created in my view an oximoron: if there is an advantage of having the LLM creating memories through a prompt, then the advantage is that this is part of its instruction set. if however, such indexing need to be done in batch over time, then (1) the advantage disappears (2) it is far less clear how these episodic memories are created, as the whole content has to enter context for Gist extraction + Fact extraction + Memory graph construction

if that's the case, then the whole construct seems a bit too artificial, as the system cannot be used to dynamically and naturally construct episodic memories (as it is presented), but can just be evaluated as an adademic toy


## lightweight contribution

irrespectively of the above conudrum, the paper contribution remains quite lightweight, as it essentiually leverages such offline-prompt-built-graph through an iterative (or signle shot, which in most cases work fine) retrieval procedure (akin to a walk into the offline-built graph)

as details of the contributions are lacking, and anyway not completely new as per authors own admiossion (e.g.,  graph is conflated with synonymous edges whose similarity increase above a threshold as in HyppoRAG), the whole construct remains quite heuristic

additionally, no analysis of the basic properties of the  constructed graph (connectivity, betweenness centrality, diameter, degree, anything) is addressed (let alone mentionoing an analysis of how such graph would differ from similar grapical abstraction of  graphRAG hyppoRAG).

with such graph information, one could study properties of the random walk (from initial random seeds) as intermediate step before reporting results of the downstream task.

lacking the above, the methodological contribution is akin to "we have engineered an heuristic"

## lack of statistical rigour in the analysis

tables report numbers that are often very close and would need a statistically relevant analysis to tell if they are apart.

for instance, my gut feeling is that Rem-I and Rem-S are statistically equivalent, i.e., a single shot suffice for most cases as the margin is too thin. this may of course depend on the hardness of the task, as intuitively a one-shot retrieval cannot be sufficient for more involved retrieval; howerver, part of the lack of advantage on Rem-I may be rooted in a poorly designed inference/retrieval strategies (as the gap from oracle is wider and so possibly statistically relevant).

the lack of statistical rigor leaves these questions unaddressed

## doubtful numbers reported, with lower than publicly reported baselines

 Mem0 reports significantly higher than 25.1 F1 score reported in Tab 3.

 Mem0 reports significantly higher than 25.1 F1 score reported in Tab 3.

		Method Single Hop Multi-Hop Open Domain Temporal
		Mem0 	38.72	 28.64 	 47.65 	 48.93
		Mem0g 	38.09	 24.32 	 49.27	 40.28

https://arxiv.org/pdf/2504.19413

---

> ### Author Response · Authors · 2025-11-23
> **To Reviewer JsEd**
>
> We thank the reviewer for recognizing the importance of the problem and the depth of our analysis, including the error breakdown and human comparison. For clarifications on the overall contribution and positioning of REMem, please see our Response to All Reviewers, and we address your specific concerns as follows.
>
> > Q1: The paper claims this indexing is still an offline phase; less clear how memories are created
>
> We agree that our “offline indexing” phase may seem restrictive, and we’ve updated the term to \`indexing\` in the paper, and will clarify this issue further.
> Our indexing phase is actually not inherently tied to a batch-only, offline workflow. What we called “offline” in the paper simply distinguishes the **backend** processing stage from the user-facing **online** inference loop.
>
> In practice, the same indexing procedure can be executed incrementally and asynchronously, exactly as in real-world memory systems (e.g., vector DB ingestion).
>
> Crucially, gist extraction and fact extraction operate at the event/session level, not over the entire history. Therefore:
>
> - Adding a new experience requires only processing that experience/session,
> - No re-ingestion of the full history is needed,
> - And updating the graph is algorithmically equivalent to what we already implement.
>
> This means the system can maintain a continuously updated memory graph in the background while the agent interacts with users, without requiring global recomputation.
> Our choice to index in a batch mode was purely for controlled and reproducible benchmarking, not a methodological limitation.
>
> > Q2: Lightweight contribution. How does it differ from GraphRAG, HippoRAG?
>
> Please refer to REMem’s Contributions in Response to All Reviewers.
> Regarding the difference from HippoRAG,
> While GraphRAG and HippoRAG also combine graphs with LLMs, they are designed as semantic knowledge retrieval systems, especially for entity-centric texts.
> For the indexing phase of REMem, the similarity to HippoRAG lies **only** in the construction of synonymous edges. For the inference phase, REMem’s capabilities in episodic reasoning are fundamentally unattainable by GraphRAG/HippoRAG, as their graph algorithms cannot execute any (temporal) logic.
>
> > Q3: Analysis of the basic properties of the constructed graph
>
> Please refer to Appendix F.5, where we have updated the paper draft about graph properties.

---

> ### Author Response · Authors · 2025-11-23
> **To Reviewer JsEd (2)**
>
> > Q4: Statistically relevant analysis to tell if REMem-I and REMem-S are apart
>
> You have keenly observed that REMem-I and REMem-S show little difference in episodic recollection tasks. While the difference is not major in Table 3, it doesn’t suggest they are statistically equivalent.
>
> 1\. On episodic reasoning tasks requiring multi-step inference, REMem-I outperformed REMem-S by 7.0 and 20.6 points on the LLM-J score for Complex-TR and Test of Time, respectively, as shown in Table 4\. This substantial improvement should not be interpreted as statistically equivalent.
>
> 2\. In LoCoMo, less than 14% of questions require multiple sessions to solve, indicating this benchmark's limited difficulty and inability to fully reflect differences in multi-step scenarios.
>
> 3\. The differences between REMem and other methods are evident. We provide 95% confidence intervals obtained via bootstrapping with 2000 resamples in the table below for reference.
>
> | Method | LoCoMo |  |  | REALTALK |  |  |
> | :---- | :---- | :---- | :---- | :---- | :---- | :---- |
> |  | F1 | BLEU-1 | LLM-J | F1 | BLEU-1 | LLM-J |
> | Qwen3-Embed-8B | 35.3 \[33.7, 37.0\] | 28.9 \[27.4, 30.7\] | 64.2 \[62.2, 66.6\] | 20.2 \[18.6, 22.0\] | 14.9 \[13.5, 16.5\] | 52.5 \[48.9, 55.9\] |
> | NV-Embed-v2 | 39.6 \[38.2, 41.3\] | 31.0 \[29.6, 32.7\] | 73.0 \[71.2, 75.0\] | 23.8 \[22.0, 25.7\] | 17.7 \[16.2, 19.2\] | 59.5 \[55.9, 62.8\] |
> | Mem0 | 25.1 \[23.6, 26.8\] | 18.0 \[16.9, 19.4\] | 49.7 \[47.5, 52.0\] | 9.8 \[8.5, 11.3\] | 7.2 \[6.2, 8.3\] | 14.3 \[12.0, 17.0\] |
> | Graphiti | 33.7 \[31.9, 35.5\] | 28.9 \[27.2, 30.8\] | 52.5 \[50.2, 54.8\] | 15.1 \[13.6, 16.9\] | 11.5 \[10.3, 12.9\] | 35.3 \[32.0, 39.0\] |
> | HippoRAG 2 | 39.0 \[37.4, 40.6\] | 30.8 \[29.3, 32.3\] | 74.0 \[71.9, 75.7\] | 21.9 \[20.3, 23.5\] | 16.2 \[14.9, 17.6\] | 55.8 \[52.2, 59.2\] |
> | REMem-I | 42.4 \[40.8, 44.0\] | 32.7 \[31.1, 34.2\] | 76.2 \[74.3, 78.2\] | 25.6 \[23.9, 27.4\] | 18.1 \[16.7, 19.7\] | 63.7 \[60.2, 67.3\] |
> | REMem-S | 41.3 \[39.8, 42.9\] | 31.5 \[30.1, 33.1\] | 77.5 \[75.9, 79.4\] | 26.2 \[24.6, 28.0\] | 19.2 \[17.9, 20.7\] | 65.3 \[62.2, 68.9\] |
>
> | Method | Complex-TR |  |  | Test-of-Time |
> | :---- | :---- | :---- | :---- | :---- |
> |  | F1 | BLEU-1 | LLM-J | EM |
> | Qwen3-Embed-8B | 77.1 \[75.0, 79.4\] | 71.4 \[69.0, 73.9\] | 80.9 \[78.4, 83.4\] | \- |
> | NV-Embed-v2 | 77.5 \[75.4, 79.7\] | 71.9 \[69.6, 74.2\] | 80.4 \[77.9, 83.0\] | 68.9 \[67.2, 70.6\] |
> | NV-Embed-v2 w/ TISER | 88.1 \[86.6, 89.8\] | 83.6 \[81.8, 85.8\] | 88.3 \[86.4, 90.2\] | 68.9 \[67.1, 70.7\] |
> | Mem0 | 43.1 \[40.4, 45.9\] | 35.1 \[32.7, 37.6\] | 41.0 \[38.0, 44.0\] | \- |
> | Graphiti | 76.6 \[74.3, 78.8\] | 71.4 \[68.9, 73.8\] | 78.8 \[76.2, 81.4\] | \- |
> | REMem-I | 83.3 \[81.5, 85.1\] | 77.6 \[75.5, 79.8\] | 89.6 \[87.6, 91.6\] | 93.1 \[92.0, 94.0\] |
> | REMem-I w/ TISER | 90.6 \[89.2, 91.8\] | 86.0 \[84.3, 87.7\] | 92.0 \[90.3, 93.6\] | 90.6 \[89.4, 91.6\] |
> | REMem-S | 78.5 \[76.4, 80.5\] | 72.7 \[70.3, 75.1\] | 82.6 \[80.2, 84.9\] | 72.5 \[70.7, 74.3\] |
>
> > Q5: Doubtful Mem0 numbers reported.
>
> First, Table 2 in Mem0 paper shows that Mem0 and Mem0\_g use 1,764 and 3,616 memory tokens on LoCoMo, respectively, without a clear explanation about how they set \`k\` and \`memory\_tokens\`. To ensure a fair comparison, we selected their top-10 retrieved sentences as the retrieval context (\<\~500 tokens), much smaller than their 1,764 tokens.
>
> Second, to align the backbone LLM and embedding model and get results on the full LoCoMo rather than its subset, we reproduce the results rather than using reported numbers. Different backbone models naturally yield variations.
>
> Third, Mem0 exists in both proprietary (https://app.mem0.ai/login) and open-source versions ([https://github.com/mem0ai/mem0](https://github.com/mem0ai/mem0)) without a technical explanation for their differences. We opted to use the latest open-source version available at the time, while this repository is also changing frequently.
>
> Finally, the open-source version of Mem0 is not entirely transparent: no script for specific benchmarks.
>
> In summary, particularly the methodological differences highlighted in the first point, multiple factors have led to this reasonable discrepancy in the reported numbers.

---

> > ### Comment · Reviewer_JsEd · 2025-11-27
> > **Noted for rebuttal**
> >
> > I think authors did a fair clarification job in the rebuttal. Some of their detailed answers, while not fully convincing, are however a step in the good direction (*, **) and others were necessary and are convincing (***). So I am willing to raise my score
> >
> >
> >
> > (*)  indexing
> > i.e., the indexing "can be" done in streaming sure, but your evaluation does it offline, in batch, only once; having previous direct experience on how things differ in practice on working from batch -> minibatch -> stream, I suspect moving to a indexing in parallel with inference will not be so smooth, but as long as it clearly clarified  )
> >
> > (**) mem0
> > at least I see the reason of the discrepancies are not yet known, but authors have done their due diligence
> >
> > (***) statistical relevance
> > please put caveats and CIs in the new version of the paper (instead of simply using bold for best, it's best if it is in a statistically meaningful way)

---

> ### Author Response · Authors · 2025-11-27
> **Thank you for raising score**
>
> Dear Reviewer JsEd,
>
> Thank you very much for your thoughtful follow-up and for raising your score. We appreciate your detailed concerns and will make sure the final version reflects them clearly.
>
> For point 1, we acknowledge that moving to streaming indexing in parallel with inference poses practical challenges. We will clarify this issue and identify areas for future work.
>
> For point 2, we will present the multiple factors we listed as possible causes and provide a more comprehensive comparative context.
>
> For point 3, we found your suggestion regarding statistical reporting very helpful. We will include CIs in the next version to qualify which improvements are statistically significant, rather than relying solely on bold text.
>
> Thank you again for your constructive feedback.

---

### Author Response · Authors · 2025-11-23
**Response to All Reviewers**

We thank all reviewers for their constructive comments, acknowledging our strong motivation, clear presentation, high-quality writing, and informative figures.
We appreciate their acknowledgment of the comprehensive empirical results on episodic recollection and reasoning (f9y5, oAJd, 7Ujh), and the depth of our analyses (JsEd, 1gLe, f9y5, oAJd, 7Ujh), including ablation (f9y5, 7Ujh), error analysis (JsEd, 1gLe, oAJd), human evaluation (oAJd, 7Ujh), comparative analysis (JsEd, 1gLe, f9y5, 7Ujh), and refusal performance exploration (f9y5, oAJd, 7Ujh).

### REMem’s Contributions to Language Agent Memory (JsEd Q2, f9y5 W2, oAJd W2)

*Recalling* and *reasoning* over past events along semantic and spatiotemporal dimensions is crucial for human productivity, yet language agent systems still struggle to mimic such episodic memory capabilities. Our work in REMem, rather than just introducing another “LLM \+ Graph” paper, aims to investigate and ultimately address this important gap in episodic memory capabilities.

We first extend the evaluation scope of existing works by defining two capabilities: **episodic recollection** (reconstructing events based on time, location, participants, and emotions) and **episodic reasoning** (performing multi-step reasoning over those events, including ordering, counting, and inter-event relations). This defined challenging scope overcame the narrow focus of prior works on conversational/semantic QA (f9y5 W3).

Given such critical challenges, we introduce the novel REMem approach, with improvements grounded in two designed dimensions: 1\) **episodic** **memory** **structuring** and 2\) **agentic** **inference**, enabling reasoning capabilities that neither embedding-based retrieval nor structured non-parametric approaches can support.

On our comprehensive evaluation across four episodic memory benchmarks, we show that REMem substantially outperforms state-of-the-art memory systems, showing 3.4% and 13.4% absolute **improvements** on episodic recollection and reasoning tasks, respectively.

**Episodic Memory Structuring: Indexing Phase (JsEd Q1, 1gLe Q8, f9y5 Q3)**

For indexing, our design transforms HippoRAG 2’s graph building by introducing spatiotemporal grounding and gist-extraction. This leads to a much more robust episodic memory representation than HippoRAG 2 and other baselines designed specifically for episodic memory, such as Mem0 and Graphiti, due to their focus on knowledge conflict and entities evolving over time rather than the spatiotemporal grounding of events (Appendix F.4).

Specifically, most non-parametric memory systems either (1) index concepts/entities, e.g., triples, KG nodes, or (2) store contextual spans. Recently, Mem0 mainly treats time as metadata on stored facts, and Graphiti as temporal annotations on fact edges; neither builds temporality into an explicit episodic layer that unifies events and their dependent facts, which is exactly what the hybrid graph of REMem is designed to do:

- Context level: gist nodes encode event-level situational frames (https://pubmed.ncbi.nlm.nih.gov/9522683/), including time, location, participants, causality, etc, especially for explicit temporal scopes (points or intervals). This preserves episodic granularity required for temporal or situational queries.
- Concept level: facts composed of (subject, predicate, object) with qualifiers enable a more precise representation of relationships between phrases.

**Agentic Inference Phase (f9y5 W1)**

We design REMem’s inference as an agentic process that operates directly on a graph-structured episodic memory representation. The agent interfaces with rich memory primitives, including semantic and lexical search, navigation over adjacent gists and facts, and temporal filtering, so that inference is data-structure-aware rather than driven only by similarity scores or graph topology.

By storing time-scoped episode gists explicitly linked to fact triples, each reasoning step can exploit the graph’s semantics, including timestamps, validity intervals, and gist–fact connections. This enables flexible episodic recollection and reasoning via the orchestration of retrieval and graph-exploration tools, a capability missing in prior paradigms:

* Embedding retrieval ranks by vector similarity, but cannot enforce multi-step constraints such as “A happened *before* B, under role R, within interval T,” nor can it validate partial hypotheses as the reasoning unfolds.
* Graph algorithms like PPR are topology-driven and semantics-blind: they explore by structural connectivity, ignoring (temporal) semantic or situational elements.

Overall, we see REMem as explicitly aligning situational context with temporal and conceptual structure for agentic reasoning. This yields a capability that prior paradigms cannot provide.

---

### Author Response · Authors · 2025-11-23
**Update Notes for Paper Draft**

The updated contents include:

- Appendix B Formal Definition: Memory structure and algorithms. (**1gLe**, **7Ujh**)
- Appendix C.7: Performance by temporal category, including non-temporal questions.  (**1gLe**, **f9y5**, **7Ujh**)
- Appendix F.3: Token usage and corresponding USD costs for both indexing and inference phases. (**1gLe**, **7Ujh**)
- Appendix F.4 Example of the Extraction (**f9y5**)
- Appendix F.5 Graph Property: Statistics of created graphs. (**JsEd**, **1gLe**, **7Ujh**)

We’ll further add more discussions and ablation studies in the revised version.

---

> ### Author Response · Authors · 2025-11-26
> **Update Notes for Paper Draft (2)**
>
> We thank the reviewers for their important suggestions. As part of our promised incremental revisions, we have updated the draft as follows:
>
> - Section 6.1 and Table 5 (Ablation Studies): We have added new ablations on synonymy edges and on different retrievers. Together with the existing ablations on gists and facts during graph construction, this yields a more complete analysis of the main components of REMem (**1gLe**, **7Ujh**).
> - Appendix F.2 (Time and Space Efficiency): We now report the time and memory costs for indexing and inference, which we hope will facilitate comparison and future analyses (**7Ujh**).
>
> We believe these updates help address the reviewers’ main concerns regarding the experiments.

---

### Meta-Review · Area_Chair_U4Xh · 2026-01-09

**Summary:**

Reviewers’ concerns primarily focused on whether the contribution was sufficiently novel beyond prior graph-based memory systems (JsEd, f9y5), the realism and practicality of the offline indexing assumption for continuously operating agents (JsEd, 7Ujh), the statistical rigor and significance of the reported gains (JsEd), and the clarity and formal specification of the memory graph and inference process (1gLe, 7Ujh). Additional concerns addressed the robustness of LLM-based extraction and the realism of time-rich benchmarks (f9y5), as well as baseline reproducibility and fairness (JsEd). These issues framed the discussion around distinguishing substantive episodic reasoning contributions from incremental engineering.

**Reviewer Concerns:**

The rebuttal and revised draft addressed most major concerns: statistical rigor was improved through confidence intervals and expanded analyses (JsEd); discrepancies with baseline results were carefully explained (JsEd); the memory graph schema, graph properties, and inference workflow were clarified with added formal descriptions and ablations (1gLe, 7Ujh); and the indexing design was clarified as incrementally applicable rather than inherently batch-only (JsEd, 7Ujh). Remaining concerns are primarily about broader generalization to more challenging real-world episodic settings with sparse or implicit temporal cues and very long contexts (f9y5), as well as long-term scalability, which are better framed as future work rather than fundamental limitations.

**Reviewer Scores:**

**Reviewer JsEd**
After the rebuttal, the reviewer indicated willingness to raise the score, conditional on adding confidence intervals and clear caveats.

**Reviewer 1gLe**
Initially cautious about contribution and methodological clarity; after revisions and added analyses, the reviewer explicitly indicated an intention to raise the contribution score.

**Reviewer f9y5**
Consistently positive about execution and presentation but remained cautious about generalization to more realistic episodic settings; the rebuttal reduced concerns but likely would not lead to a major score increase.

**Reviewer oAJd**
Moderately positive throughout, with no major unresolved concerns; the score would likely remain unchanged.

**Reviewer 7Ujh**
Initially concerned about formalism and scalability; after the rebuttal and revisions, the reviewer explicitly stated willingness to substantially raise the score.

---

### Decision · Program_Chairs · 2026-01-26

Accept (Poster)